# MODELING EXPERT INTERACTIONS IN SPARSE MIXTURE OF EXPERTS VIA GRAPH STRUCTURES

## ABSTRACT

Sparse Mixture of Experts (SMoE) has emerged as a promising solution to achieving unparalleled scalability in deep learning by decoupling model parameter count from computational cost. By activating only a small subset of parameters per sample, SMoE enables significant growth in model capacity while maintaining efficiency. However, SMoE struggles to adapt to distributional shifts, leading to reduced robustness under data contamination. In this work, we introduce SymphonySMoE, a novel family of SMoE that introduces a social graph to model interactions among experts. This graph-based structure enhances the token routing process, addressing the robustness challenges that are inherent in conventional SMoE designs. SymphonySMoE is lightweight, modular, and integrates seamlessly with existing SMoE-based models such as the XMoE and the Generalist Language Model. We provide both theoretical analysis and empirical evidence demonstrating SymphonySMoE's advantages over baseline SMoE. Extensive experiments on language modeling and visual instruction tuning validate our method's effectiveness. We further highlight the scalability of SymphonySMoE to models with 4.2 and 7.4 billion parameters, showcasing its applicability in fine-tuning tasks for large-scale systems.

## 1 INTRODUCTION

Scaling deep models has emerged as a powerful paradigm for enhancing performance across a wide range of cognitive and machine learning tasks, including large language model pre-training (Devlin et al., 2019; Radford et al., 2019; Raffel et al., 2020; Kaplan et al., 2020; Brown et al., 2020; Nguyen et al., 2022; Touvron et al., 2023), vision understanding (Dosovitskiy et al., 2021; Bao et al., 2022a;b; Li et al., 2023a; Bai et al., 2024; Liu et al., 2023a), reinforcement learning (Chen et al., 2021; Janner et al., 2021), and scientific applications (Subramanian et al., 2024; Yang et al., 2023). However, this scaling paradigm comes with substantial computational demands, often presenting significant challenges in terms of efficiency and scalability. To mitigate these constraints, Sparse Mixture of Experts (SMoE) has emerged as a promising strategy for efficiently scaling deep models. By structuring the model into modular components and selectively activating only a subset of experts per input, SMoE achieves a trade-off between computational efficiency and increased model capacity. This approach has enabled the development of billion-parameter models that achieve state-of-the-art performance in tasks such as machine translation (Lepikhin et al., 2021), image classification (Riquelme et al., 2021), and speech recognition (Kumatani et al., 2021), all while maintaining a constant computational footprint.

### 1.1 SPARSE MIXTURE OF EXPERTS

An MoE substitutes a component within a model layer, such as a feed-forward or convolutional layer, with a collection of specialized networks known as experts. While this design significantly enhances model scalability, it also leads to higher computational costs. In particular, an MoE consists of a router and $M$ expert networks, $u_j$, $j = 1, 2, \ldots, M$. For each input token $\boldsymbol{x}_i \in \mathbb{R}^D$, $i = 1, 2, \ldots, N$, the MoE's router computes the router scores between $\boldsymbol{x}_i$ and each expert as $\gamma_j(\boldsymbol{x}_i)$, $j = 1, 2, \ldots, M$. In practice, we often choose the router $\gamma(\boldsymbol{x}_i) = [\gamma_1(\boldsymbol{x}_i), \gamma_2(\boldsymbol{x}_i), \ldots, \gamma_M(\boldsymbol{x}_i)]^\top = \boldsymbol{W}\boldsymbol{x}_i + \boldsymbol{b}$, where the router weight $\boldsymbol{W} := [\boldsymbol{w}_1, \ldots, \boldsymbol{w}_M]^\top \in \mathbb{R}^{M \times D}$ is the matrix composed of expert embeddings $\{\boldsymbol{w}_j\}_{j=1}^M$ and the router bias $\boldsymbol{b} = [b_1, \ldots, b_M]^\top \in \mathbb{R}^M$. The outputs from all $M$ experts are then

linearly combined as

$$\boldsymbol{y}_i = \sum_{j=1}^{M} \underbrace{\text{softmax}(\gamma_j(\boldsymbol{x}_i))}_{g_j(\boldsymbol{x}_i)} u_j(\boldsymbol{x}_i), \tag{1}$$

where $\text{softmax}(\gamma_i) := \exp(\gamma_i)/\sum_{j=1}^{M}\exp(\gamma_j)$ and the expert output $u_j(\boldsymbol{x}_i) \in \mathbb{R}^D$. Here, $g_j(\boldsymbol{x}_i)$, $j = 1, \ldots, M$ are the gate values corresponding to the input token $\boldsymbol{x}_i$. Note that we distinguish between these gate values $g_j$ and the router scores $\gamma_j$: $\gamma_j(\boldsymbol{x}) = \boldsymbol{w}_j^\top \boldsymbol{x} + b_j$ and $g_j(\boldsymbol{x}) = \text{softmax}(\gamma_j(\boldsymbol{x}))$.

An SMoE inherits the extended model capacity from MoE but preserves the computational overhead by taking advantage of conditional computation. In an SMoE, a sparse gating function $\text{TopK}$ is applied to select only $K$ experts with the greatest router scores (Shazeer et al., 2017b). Here, we define the $\text{TopK}$ function as:

$$\text{TopK}(\gamma_j) := \begin{cases} \gamma_j, & \text{if } \gamma_j \text{ is in the } K \text{ largest elements of } \gamma \\ -\infty, & \text{otherwise.} \end{cases}$$

The outputs from an SMoE are then computed as

$$\boldsymbol{y}_i = \sum_{j=1}^{M} \text{softmax}(\text{TopK}(\gamma_j(\boldsymbol{x}_i))u_j(\boldsymbol{x}_i). \tag{2}$$

Alternatively, like in Switch Transformer, which integrates SMoE into transformer architectures (Fedus et al., 2022), the sparse gating function $\text{TopK}$ can be used to select $K$ experts with the greatest gate values. In this case, the outputs from an SMoE are given by:

$$\boldsymbol{y}_i = \sum_{j=1}^{M} \text{TopK}(\text{softmax}(\gamma_j(\boldsymbol{x}_i))u_j(\boldsymbol{x}_i). \tag{3}$$

We often set $K = 2$, i.e., top-2 routing, as this configuration has been shown to provide the best trade-off between training efficiency and testing performance (Lepikhin et al., 2021; Du et al., 2022; Zhou et al., 2023).

**Limitations of SMoE.** Despite its success, SMoE struggles to adapt to distributional shifts, which weakens its robustness under data contamination (Puigcerver et al., 2022; Zhang et al., 2023) and constrains its applicability in many large-scale, real-world tasks.

### 1.2 Contribution

In this paper, we study a probabilistic graphical model (PGM) framework to analyze token routing in (S)MoE. Building on this foundation, we introduce a novel PGM that explicitly captures expert-to-expert relationships in SMoE. By estimating gate values as posterior probabilities within this framework, we derive the Symphony Sparse Mixture of Experts (SymphonySMoE)–a model that enhances token routing by leveraging the social graph between experts, enabling coordinated behavior of the experts through structured interactions, much like instruments harmonizing within an orchestra (see Figure 2 in Appendix A). Our contribution is three-fold:

1. We propose a principled method to construct the social graph that represents relationships between experts.

2. We invent SymphonySMoE, an innovative model that utilizes the expert social graph to improve token routing.

3. We provide rigorous theoretical analysis demonstrating that SymphonySMoE reinforces expert selection with the largest regions of ideal co-selection, maintaining robustness even in the presence of data contamination.

Our experimental results validate that our SymphonySMoE improves over the baseline SMoE in terms of accuracy and robustness on a variety of practical benchmarks, including WikiText-103 language modeling and Llava visual instruction tuning. We also empirically demonstrate that our SymphonySMoE is universally applicable to many existing SMoE models, including the XMoE (Chi et al., 2022) and the Generalist Language Model (GLaM) (Du et al., 2022). Furthermore, we

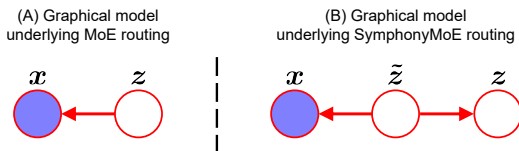

Figure 1: Graphical model underlying (A) MoE routing and (B) SymphonyMoE routing.

demonstrate the scalability of SymphonySMoE by incorporating it into SMoE models with 4.2 and 7.4 billion parameters, showcasing its applicability in fine-tuning tasks for large-scale systems.

**Organization.** We structure this paper as follows: In Section 2, we discuss (S)MoE from the conditional mixture model and PGM perspective. We then introduce a novel PGM that captures expert relationships and use it to develop our proposed SymphonySMoE model. In Section 3, we provide theoretical guarantees for the co-selection of robust expert pairings in SymphonySMoE. In Section 4, we present our experimental results to justify the advantages of our SymphonySMoE models over the traditional SMoE. In Section 5, we empirically analyze our SymphonySMoE. We discuss related works in Section 6. The paper ends with concluding remarks.

## 2 SYMPHONY SPARSE MIXTURE OF EXPERTS

### 2.1 REVIEW: GATE VALUES IN MoE AS POSTERIOR DISTRIBUTION

Considering an input token $\boldsymbol{x} \in \mathbb{R}^D$ and its corresponding output $\boldsymbol{y} \in \mathbb{R}^D$, we first review the reformulation of MoE as a conditional mixture model in (Jacobs et al., 1991; Xu et al., 1994):

$$p(\boldsymbol{y}|\boldsymbol{x}, \boldsymbol{\theta}) = \sum_{j=1}^{M} g_j(\boldsymbol{x}; \boldsymbol{W}) p(\boldsymbol{y}|\boldsymbol{x}, \boldsymbol{\theta}_j), \tag{4}$$

where $g_j$, $j = 1, \ldots, M$, are again the gate values, and $\boldsymbol{\theta}$ consists of the expert embeddings $\boldsymbol{W}$ and the expert parameters $\{\boldsymbol{\theta}_j\}_{j=1}^{M}$. Here, $p(\boldsymbol{y}|\boldsymbol{x}, \boldsymbol{\theta}_j)$ are density functions from the exponential family such as the Gaussian. The outputs from the experts $u_j(\boldsymbol{x})$, $j = 1, \ldots, M$, in Eqn. 1 contribute to the natural parameters of $p(\boldsymbol{y}|\boldsymbol{x}, \boldsymbol{\theta}_j)$, such as the mean of the distribution when $p(\boldsymbol{y}|\boldsymbol{x}, \boldsymbol{\theta}_j)$ is a Gaussian.

Let us introduce an $M$-dimensional binary random variable $\boldsymbol{z}$ having a 1-of-$M$ representation in which a particular element $z_j$ is equal to 1 and all other elements are equal to 0. We use $z_j$ to indicate the index $j$ of the expert $u_j$, i.e., the expert $u_j$ is selected if $z_j = 1$; otherwise, it is not selected. Following (Xu et al., 1994), we formulate the gate values $g_j$ as the posterior probability $p(z_j = 1|\boldsymbol{x})$ that $\boldsymbol{x}$ is assigned to the expert $\boldsymbol{u}_j$. In particular, we consider the graphical representation of the mixture model for the token $\boldsymbol{x}$ with corresponding latent variable $\boldsymbol{z}$ in Figure 1A, in which the joint distribution is expressed in the form $p(\boldsymbol{x}, \boldsymbol{z}) = p(\boldsymbol{z})p(\boldsymbol{x}|\boldsymbol{z})$. We further assume that $p(\boldsymbol{x}|z_j = 1) = \mathcal{N}(\boldsymbol{x}|\boldsymbol{w}_j, \sigma_j^2 \mathbf{I})$, representing a Gaussian distribution, and denote the prior probability $p(z_j = 1)$ as $\pi_j$. The posterior probability that token $\boldsymbol{x}$ selects expert $u_j$, i.e, the gate values $g_j(\boldsymbol{x}; \boldsymbol{W})$, is then expressed as:

$$g_j(\boldsymbol{x}; \boldsymbol{W}) = p(z_j = 1|\boldsymbol{x})$$
$$= \frac{\pi_j \mathcal{N}(\boldsymbol{x}|\boldsymbol{w}_j, \sigma_j^2 \mathbf{I})}{\sum_{j'=1}^{K} \pi_{j'} \mathcal{N}(\boldsymbol{x}|\boldsymbol{w}_{j'}, \sigma_{j'}^2 \mathbf{I})} = \frac{\pi_j \exp\left[(\boldsymbol{w}_j^\top \boldsymbol{x} + b_j)/\sigma_j^2\right]}{\sum_{j'=1}^{K} \pi_{j'} \exp\left[(\boldsymbol{w}_{j'}^\top \boldsymbol{x} + b_{j'})/\sigma_{j'}^2\right]}, \tag{5}$$

where $b_j = -\|\boldsymbol{w}_j\|^2/2$ since the input-dependent term $-\|\boldsymbol{x}\|^2/2$ cancels out in softmax normalization. Assuming the prior $\pi_j$ is uniform and choosing $\sigma_j^2 = 1$, we demonstrate that the right-hand side of Eqn. 5 matches the gate values in an MoE as in Eqn. 1, in which $\boldsymbol{w}_j$ and $b_j$ in Eqn. 5 correspond to the expert embedding and bias in the MoE defined in Eqn. 1.

**Gate Values in SMoE as Truncated Posterior.** Applying a sparse gating function TopK to select top $K$ experts with the greatest router scores as in Eqn. 2 is equivalent to using a truncated approximation to the posterior $p(\boldsymbol{z}|\boldsymbol{x})$ as in (Sheikh et al., 2014). The truncated approximation is defined to be proportional to the true posteriors on subspaces of the latent space with high probability mass. This truncated approach can be regarded as a variational approach with truncated posteriors as variational distributions.

**Remark 1** (Deriving Other Routers)**.** *The formulation of gate values in SMoE as a posterior distribution can be trivially applied to derive other popular routing mechanisms for SMoE, including the cosine router (Chi et al., 2022) or random router (Chen et al., 2023) (See Appendix C).*

---

**Algorithm 1** Symphony Routing in Sparse Mixture of Experts

---

1: **Initialize:** adjacency $A \in \mathbb{R}^{M \times M}$ as zeros
2: **Input:** tokens $\{x_i\}_{i=1}^N$, original gating function $\gamma(\cdot) : \mathbb{R}^d \to \mathbb{R}^M$, number of experts $M$, number of selected experts $K$, moving average coefficient $\beta$
3: $A' \leftarrow 0_{M \times M}$
4: **for** $i = 1, \ldots, N$ **do**
5:    $g_i \leftarrow \gamma(x_i)$                                          ▷ Compute gating logits
6:    $\mathcal{I} \leftarrow \text{TopK}(g_i, K)$
7:    $(s_i)_j \leftarrow 1$ for $j \in \mathcal{I}$
8:    $A' \leftarrow A' + s_i s_i^\top$                                  ▷ Count co-selections
9: **end for**
10: **for** $i = 1, \ldots, N$ **do**
11:    $g_i^{\text{symphony}} \leftarrow A \, \text{softmax}(g_i)$                    ▷ Smooth gating with adjacency
12:    $\mathcal{I} \leftarrow \text{TopK}(g_i^{\text{symphony}}, K)$
13:    $y_i \leftarrow \sum_{j \in \mathcal{I}} g_{ij}^{\text{symphony}} u_j(x_i)$
14: **end for**
15: $A' \leftarrow \text{RowNorm}(A')$                             ▷ Normalize adjacency
16: $A \leftarrow \beta A + (1 - \beta) A'$                             ▷ EMA update

---

**Remark 2** ((S)MoE as a Social Recommender System). *Considering the gate values in (S)MoE as a posterior distribution, the process of routing a token $\boldsymbol{x}_i$, $i = 1, \ldots, N$, to the right experts in (S)MoE can be interpreted as solving a recommendation problem (Sharma et al., 2024; Wu et al., 2022). Specifically, this involves predicting the rating an expert would assign to the token, based on the router scores $\gamma_j(\boldsymbol{x}_i)$ that represent the relationship between token $\boldsymbol{x}_i$ and expert $u_j$, $j = 1, \ldots, M$. These router scores form an expert-token matrix, equivalent to the user-item rating matrix in a social recommendation system.*

### 2.2 SYMPHONYSMOE: TOKEN ROUTING WITH SOCIAL GRAPH

The perspective of (S)MoE as a social recommender system, as discussed in Remark 2, highlights a potential limitation in the current (S)MoE setup: the relationships between experts are overlooked. According to social influence theory in the context of social recommender systems, connected users influence one another (Wu et al., 2022). Inspired by this observation, we revise the graphical model of MoE in Figure 1A to incorporate relationships between experts and propose the novel SymphonySMoE, which leverages expert-to-expert interactions to enhance its token routing.

**Modeling the Expert-to-Expert Relation.** In order to model the relationship between experts, we introduce another binary random variable, $\tilde{\boldsymbol{z}}$, into the graphical model of MoE in Figure 1A. $\tilde{\boldsymbol{z}}$ is a replica of the latent variable $\boldsymbol{z}$ that indicates the expert selection, i.e., the expert $u_j$ is selected if $\tilde{z}_j = 1$; otherwise, it is not selected. The new graphical model is depicted in Figure 1B. The relationship between the experts is captured by the conditional distribution $p(\boldsymbol{z}|\tilde{\boldsymbol{z}})$.

**Inferring the Selected Expert.** The posterior probability that token $\boldsymbol{x}$ selects expert $u_j$, i.e. the gate values $g_j^{\text{symphony}}(\boldsymbol{x}; \boldsymbol{W})$, is given by:

$$g_j^{\text{symphony}}(\boldsymbol{x}; \boldsymbol{W}) = p(z_j = 1|\boldsymbol{x}) = \sum_{k=1}^M p(z_j = 1, \tilde{z}_k = 1|\boldsymbol{x})$$

$$= \sum_{k=1}^M p(z_j = 1|\tilde{z}_k = 1) p(\tilde{z}_k = 1|\boldsymbol{x}) = \sum_{k=1}^M a_{jk} g_k(\boldsymbol{x}; \boldsymbol{W}). \qquad (6)$$

Here, we denote the conditional probability $p(z_j = 1|\tilde{z}_k = 1)$ as $a_{jk}$, and $g_k(\boldsymbol{x}; \boldsymbol{W}) = p(\tilde{z}_k = 1|\boldsymbol{x})$ since $\tilde{\boldsymbol{z}}$ is a replica of $\boldsymbol{z}$. In Eqn. 6 above, the conditional probability $a_{jk} = p(z_j = 1|\tilde{z}_k = 1)$ models the expert-to-expert relation. Eqn. 6 implies that the gate value of expert $u_k$ given the input token $\boldsymbol{x}$ influences the gate value of expert $u_j$ if there is a connection between these two experts.

**Constructing the Social Graph between Experts.** The conditional probabilities $a_{jk}$, $j, k = 1, \ldots, M$ form the adjacency matrix $\boldsymbol{A}$ of the social graph $\mathcal{S}$ between experts in an SMoE with

$\boldsymbol{A}(j, k) = a_{jk}$. We construct the social graph $\mathcal{S}$ by estimating $a_{jk}$ as follows:

$$a_{jk} = p(z_j = 1|\tilde{z}_k = 1) = \frac{p(z_j = 1, \tilde{z}_k = 1)}{p(\tilde{z}_k = 1)} = \frac{\int_{\boldsymbol{x}} p(z_j = 1, \tilde{z}_k = 1, \boldsymbol{x})d\boldsymbol{x}}{p(\tilde{z}_k = 1)}$$

$$= \frac{1}{\pi_k} \int_{\boldsymbol{x}} p(z_j = 1, \tilde{z}_k = 1|\boldsymbol{x})p(\boldsymbol{x})d\boldsymbol{x} = \frac{1}{\pi_k N} \sum_{i=1}^{N} p(z_j = 1, \tilde{z}_k = 1|\boldsymbol{x}_i). \tag{7}$$

Eqn. 7 suggests that $a_{jk} \propto \frac{1}{N} \sum_{i=1}^{N} p(z_j = 1, \tilde{z}_k = 1|\boldsymbol{x}_i)$ and can be computed by counting how many times the expert $j$ and $k$ are selected by the input tokens and then normalizing the result. Notice that $a_{jk} = a_{kj}$, and thus, the adjacency matrix $\boldsymbol{A}$ of the social graph $\mathcal{S}$ is symmetric. Also, the social graph $\mathcal{S}$ in this case is a weighted graph. We summarize our algorithm to construct the social graph $\mathcal{S}$ between experts in Algorithm 1. Note that the computation of the base gating score $\gamma(\cdot)$ in Line 5 of Algorithm 1 can follow any formulation used in other SMoE variants, such as XMoE (Chi et al., 2022) or optimal transport gating (Tianlin Liu, 2023). This design choice makes our algorithm modular and allows it to be seamlessly integrated into existing SMoE architectures.

**Remark 3** (Connection of Algorithm 1 and the Hebbian Learning Rule). *We can interpret the update rule for our adjacency matrix as a special case of the Hebbian style learning rule, which strengthens connections between expert nodes if they fire together (Hebb, 2005; Hopfield, 1982).*

We now define our SymphonySMoE.

**Definition 1** (Symphony Sparse Mixture of Experts). *A SymphonySMoE is a Sparse Mixture of Experts that incorporates the adjacency matrix $\boldsymbol{A} = (a_{jk})_{1 \leq j,k \leq M}$ of the social graph $\mathcal{S}$ between experts constructed by Algorithm 1 into its token routing mechanism to compute the output tokens $\boldsymbol{y}_i$, $i = 1, \ldots, N$, as follows:*

$$\boldsymbol{y}_i = \sum_{j=1}^{M} \text{TopK} \left( \sum_{k=1}^{M} a_{jk} softmax(\gamma_k(\boldsymbol{x}_i)) \right) u_j(\boldsymbol{x}_i), \tag{8}$$

*where $\gamma_k(\boldsymbol{x}_i) = \boldsymbol{w}_k^\top \boldsymbol{x}_i + b_k$, $k = 1, \ldots, M$ and $i = 1, \ldots, N$, are the router scores between the experts $u_k$ and the input tokens $\boldsymbol{x}_i$ as defined in Section 1.1.*

**Remark 4** (Symphony Router). *The gate values in SymphonySMoE are given as $g_j^{symphony}(\boldsymbol{x}_i) = \sum_{k=1}^{M} a_{jk} softmax(\gamma_k(\boldsymbol{x}_i))$, and we define the Symphony Router as the router whose gate values corresponding to the input token $\boldsymbol{x}_i$ are $g_j^{symphony}(\boldsymbol{x}_i)$, $i = 1, \ldots, N$ and $j = 1, \ldots, M$.*

**Remark 5.** *Our Symphony Router and SymphonySMoE adopt the setting in Eqn. 3 that uses the sparse gating function TopK to select $K$ experts with the greatest gate values as in Switch Transformer (Fedus et al., 2022).*

## 3 THEORETICAL ANALYSIS ON CO-SELECTION OF HIGH-CONFIDENCE EXPERT PAIRINGS

Next, we provide a clear interpretation of the estimated coefficients $a_{jk}$ and demonstrate how they naturally reinforce the selection of expert pairs with the largest confidence regions of ideal co-selection under mild theoretical assumptions. First, in Theorem 1, we show that $a_{jk}$ reliably estimates the measure of the confidence regions in which both experts $i$ and $j$ should be activated. Then, in the following remarks, we explain how this theoretical property improves the accuracy and stability of the routing mechanism by promoting the co-selection of high-confidence pairs for the given input token.

**Theorem 1.** *Let $\varepsilon > 0$ be a small number and suppose that the input tokens are contaminated with independent additive noise $\boldsymbol{\delta}$ such that $\|\boldsymbol{\delta}\| \leq \varepsilon$. Let $\mathcal{C}_{jk}$ be the region such that the optimal expert selections for the tokens in $\mathcal{C}_{jk}$ are experts $j$ and $k$, and define $\gamma_{N,\varepsilon}(\alpha) := \sqrt{\frac{\ln(2/\alpha)}{2N}} + \tilde{L}\varepsilon$ with a constant $\tilde{L}$ that depends only on the input space. Then, the bound*

$$|a_{jk} - \mu(\mathcal{C}_{jk})| \leq \gamma_{N,\varepsilon}(\alpha)$$

*holds with probability at least $1 - \alpha$, where $\mu(\cdot)$ is an appropriate probability measure.*

Table 1: Perplexities (PPL) of SymphonySMoE, SymphonyGLaM, and SymphonyXMoE vs. the corresponding SMoE baselines on clean/attacked WikiText-103 validation/test sets. A lower PPL implies better performance.

| Model/Metric | Parameters | Clean WikiText-103 | | Attacked WikiText-103 | |
|---|---|---|---|---|---|
| | | Valid PPL ↓ | Test PPL ↓ | Valid PPL ↓ | Test PPL ↓ |
| *SMoE (baseline)* | 216 M | 33.76 | 35.55 | 42.24 | 44.19 |
| SymphonySMoE (Ours) | 216 M | **32.71** | **34.29** | **40.87** | **42.79** |
| *GLaM (baseline)* | 220 M | 36.37 | 37.71 | 45.83 | 47.61 |
| SymphonyGLaM (Ours) | 220 M | **36.35** | **37.60** | **45.00** | **46.43** |
| *XMoE (baseline)* | 216 M | 33.21 | 34.59 | 44.27 | 45.68 |
| SymphonyXMoE (Ours) | 216 M | **33.07** | **34.51** | **41.26** | **42.68** |

Theorem 1 ensures that $a_{jk}$ concentrates on expert pairs with the highest co-selection probabilities $\mu(\mathcal{C}_{jk})$. As a result, $g^{\text{symphony}}$ is downweighted for experts with poor average compatibility with other experts for the given input, leading to more accurate predictions. Note that additive noise is considered to model the natural, non-deterministic data distributions and contamination. The proof is provided in Appendix B.1.

**Remark 6** (Consistency). *If there is no contamination and tokens are distributed in alignment with ideal co-selection regions, the result implies that $a_{jk}$ is a consistent estimator of $\mu(\mathcal{C}_{jk})$ since $\gamma_{N,\varepsilon}(\alpha) = O(N^{-1/2} + \varepsilon)$ ensures that*

$$|a_{jk} - \mu(\mathcal{C}_{jk})| \le \gamma_{N,\varepsilon}(\alpha) \xrightarrow[\varepsilon \to 0]{} \sqrt{\frac{\ln(2/\alpha)}{2N}} \xrightarrow[N \to \infty]{} 0.$$

**Remark 7** (Promotion of High-Confidence Pairings). *By redefining the routing scores in $g_j^{symphony}(\boldsymbol{x}; \boldsymbol{W})$, the router learns to promote the choice of experts such that their compatibility with other experts when co-activated is expected to be relatively higher, quantified by the combined measures of ideal co-selection regions $\mathcal{C}_{jk}$ as in Theorem 1. Likewise, it discourages the co-selection of experts when their shared activation regions exhibit a higher density of contamination-sensitive tokens—such as tokens concentrated near the lower-confidence boundary of the activation region $\mathcal{C}_{jk}$ in most cases. This leads to an underestimation of $\mu(\mathcal{C}_{jk})$ (smaller $a_{jk}$), as any contamination or perturbation increases the likelihood of tokens near the boundary drifting out of $\mathcal{C}_{jk}$.*

The following proposition makes the effect of $A$ on the gating probability distribution explicit by showing that it contracts mean-zero perturbations and raises the margin required for an adversarial TopK change. The proof is deferred to Appendix B.3.

**Proposition 1.** *Let $A \in \mathbb{R}^{M \times M}$ be the adjacency matrix defined by $a_{jk} = P(z_j = 1 \mid \tilde{z}_k = 1)$. By construction, $A$ is a nonnegative, symmetric, and doubly-stochastic matrix. Assume the undirected graph induced by $A$ is connected and aperiodic, so that the eigenvalues of $A$ satisfy $1 = \lambda_1 > \max_{i \ge 2} |\lambda_i| =: \rho < 1$. Then for any gating probability vector $s \in \Delta^{M-1}$ and $r := As$:*

(i) ***Contraction on mean-zero perturbations:*** *For $v$ with $\mathbf{1}^\top v = 0$, $\|Av\|_2 \le \rho\|v\|_2$, so $\|A(s' - s)\|_2 \le \rho\|s' - s\|_2$ for probability vectors $s, s'$.*

(ii) ***TopK stability:*** *Let $J$ be the TopK indices of $r$, with margin $g = \min_{j \in J} r_j - \max_{j \notin J} r_j > 0$. If $\|\Delta s\|_2 < g/(2\rho)$, the TopK set of $r' = A(s + \Delta s)$ is $J$, with input bound $\|\Delta x\| < g/(2\rho L_s)$ if $s(x)$ is $L_s$-Lipschitz.*

## 4 EXPERIMENTAL RESULTS

In this section, we empirically verify the advantages of our SymphonySMoE over the baseline SMoE on WikiText-103 language modeling (Merity et al., 2017), Visual Instruction Tuning tasks (Liu et al., 2024d), and GLUE finetuning (Wang et al., 2018). We aim to show that: (i) SymphonySMoE enhances model performance across both pre-training and fine-tuning tasks; (ii) SymphonySMoE is highly effective in stabilizing the model's router against data corruption, leading to increased robustness; (iii) SymphonySMoE can be seamlessly integrated into a wide range of SMoE architectures; (iv) SymphonySMoE is scalable and continues to be effective in large-scale models.

Throughout our experiments, we replace the routing mechanism in each SMoE baseline model with our new Symphony Router (Remark 4) and compare their performance on clean and corrupted data

Table 2: Accuracy of SMoE and SymphonySMoE in the Visual Instruction Tuning task. Both models are upcycled from Siglip224 + Phi3.5 with a total of 4.2B parameters and fine-tuned on LLaVA-665K (Liu et al., 2024c) dataset. We evaluate the model performance across 7 popular benchmarks with diverse characteristics, especially in hallucination with POPE (Li et al., 2023b) and robustness with MMBench (Liu et al., 2024e). A higher accuracy indicates better performance.

| Benchmark/Model | TextVQA | GQA | MMMU | MMStar | ScienceQA | MMBench-EN | POPE |
|---|---|---|---|---|---|---|---|
| *SMoE (baseline)* | 41.27 ±0.08 | 60.90 ±0.06 | 41.78 ±0.17 | 41.41 ±0.20 | 79.33 ±0.28 | 70.66 ±0.18 | 85.22 ±0.04 |
| SymphonySMoE (Ours) | **41.57** ±0.04 | **61.20** ±0.02 | **42.56** ±0.06 | **42.44** ±0.14 | **81.87** ±0.17 | **71.74** ±0.16 | **86.84** ±0.03 |

for pre-training tasks and during fine-tuning. More experimental results and details on datasets, models, and training are provided in Appendix D and Appendix E.

### 4.1 WikiText-103 Language Modeling

**Experimental Setup.** We compare our SymphonySMoE with the Switch Transformer (SMoE in Table 1), GLaM (Du et al., 2022), and XMoE (Chi et al., 2022) with 16 experts per SMoE layer in the medium configuration, using top-2 expert routing. We present the pre-training validation and test perplexity (PPL) results for WikiText-103 in Table 1.

**Robust Language Modeling.** In Table 1, we further report the validation and test PPL on an attacked WikiText-103 dataset. We contaminate the WikiText-103 dataset using the word-swap attack from TextAttack (Morris et al., 2020) and randomly replace words with the generic AAA token. We follow the setup of (Han et al., 2024) and assess models by training them on clean data and evaluating them on the corrupted dataset.

We observe consistent improvements across all metrics and baselines, with particularly larger gains on the attacked dataset. Our results validate the effectiveness of the proposed Symphony Router, demonstrating not only improved performance on clean data but also enhanced robustness. Moreover, the consistent gains observed in vanilla SMoE, GLaM, and XMoE highlight its broad applicability across diverse model architectures.

### 4.2 Visual Instruction Tuning: Large-Scale Experiments with a 4.2-Billion-Parameter SMoE

**Experimental Setup.** This experiment addresses the Visual Instruction Tuning task (Liu et al., 2024d), following the methodology of CUMO (Li et al., 2024) to upcycle a dense Visual Language Model (VLM) into its Mixture of Experts (MoE) counterpart. We employ the pretrained SigLIP 224 + Phi3.5 as the dense model and upcycle it into a total of 4 experts, utilizing a top-2 expert routing strategy. The resulting MoE model is notably large-scale, with 4.2 billion parameters, making it a practical and powerful candidate for real-world applications. Further details regarding the training setup are provided in Appendix D.3.

**Evaluation benchmarks.** We evaluate the fine-tuned model on 7 widely recognized benchmarks for Visual Language Models (VLMs) (Liu et al., 2024c; Li et al., 2024; Wang et al., 2024b). The first five benchmarks–GQA (Hudson & Manning, 2019), TextVQA (Singh et al., 2019), MMMU (Yue et al., 2024), MMStar (Chen et al., 2024), and ScienceQA (Lu et al., 2022)–assess the model's visual and textual capabilities across diverse scenarios. Additionally, MMBench (Liu et al., 2024e) evaluates the model's answer robustness through comprehensive shuffling of multiple-choice answers. Lastly, POPE (Li et al., 2023b) measures object hallucination in images using three sampled subsets of COCO: random, common, and adversarial.

**Robust Large Multimodal Model.** As shown in Table 2, SymphonySMoE demonstrates consistent and significant improvements over the vanilla SMoE across all benchmarks, underscoring the effectiveness of integrating the social graph into the router. Notably, SymphonySMoE achieves a 1.08% improvement on the MMBench-EN benchmark and a 1.62% improvement on the POPE benchmark, highlighting its enhanced robustness in both textual and visual understanding. Across the five remaining core benchmarks–TextVQA, GQA, MMMU, MMStar, and ScienceQA–SymphonySMoE consistently outperforms the baseline SMoE, with particularly notable gains on ScienceQA, where it achieves an improvement of 2.54%. These results demonstrate the practical advantages of SymphonySMoE, making it a more reliable and effective model for real-world applications.

### 4.3 GLUE text classification on 7.4-Billion-Parameter SMoE

**Experimental Setup.** To further assess scalability, we fine-tune a sparsely upcycled Phi3-mini model (Abdin et al., 2024) on GLUE (Wang et al., 2018), where each MLP is replaced with a 4-expert

Table 3: Performance of Phi3-SMoE variants on 8 fine-tuning tasks for GLUE. All SMoE variants select top-2 among 4 experts. Across all metrics, higher scores indicate better performance.

| Methods | Params | STSB | MRPC | CoLA | RTE | QQP | QNLI | SST2 | WNLI |
|---|---|---|---|---|---|---|---|---|---|
| SMoE Phi3 | 7.4B | 87.05 | 90.37 | 60.81 | 84.48 | 92.14 | 94.62 | 95.64 | 61.97 |
| SymphonySMoE Phi3 | 7.4B | **87.92** | **91.53** | **62.46** | **85.56** | **92.26** | **94.93** | **96.22** | **66.20** |

Table 4: Perplexity comparison of SymphonySMoE and SMoE with different expert configurations on clean and attacked WikiText-103 validation and test sets. A lower PPL implies better performance.

| Model/Metric | Clean WikiText-103 | | Attacked WikiText-103 | |
|---|---|---|---|---|
| | Valid PPL ↓ | Test PPL ↓ | Valid PPL ↓ | Test PPL ↓ |
| *SMoE-shared (A)* | 32.67 | 34.12 | 41.18 | 42.75 |
| SymphonySMoE-shared (A) | **31.91** | **33.39** | **40.23** | **41.92** |
| *SMoE-32-top4 (B)* | 30.77 | 32.71 | 43.77 | 45.72 |
| SymphonySMoE-32-top4 (B) | **30.54** | **32.01** | **38.29** | **40.07** |

MoE (top-2 routing) totaling 7.4B parameters. This setup reflects a realistic deployment scenario where balancing accuracy and efficiency is essential. Fine-tuning uses standardized hyperparameters across GLUE tasks for fair comparison with the baseline SMoE.

**Evaluation Benchmark.** We evaluate on eight GLUE tasks (STSB, MRPC, QQP, CoLA, RTE, QNLI, WNLI, SST2), spanning sentence-level and pairwise classification, with performance reported using the official GLUE evaluation metrics.

**Consistent Improvement across All Tasks.** Table 3 shows that SymphonySMoE delivers consistent gains over the baseline SMoE implementation across all eight GLUE tasks under the same parameter budget. The improvements hold for both high-performing tasks such as QNLI (94.93 vs. 94.62) and SST2 (96.22 vs. 95.64), as well as more challenging ones such as WNLI (66.20 vs. 61.97). This uniform improvement pattern indicates that SymphonySMoE makes more effective use of expert capacity, reducing performance variance across tasks. Overall, these results confirm that SymphonyS-MoE consistently enhances task accuracy in large-scale configurations without compromising the scalability benefits of sparse expert activation.

### 4.4 ADDITIONAL EXPERIMENT RESULTS

Beyond the main text experiments, Appendix E presents extended results showing SymphonySMoE's broad applicability. On language modeling (C4-subset), it lowers perplexity under both clean and attacked settings. In vision, SymphonySwin-MoE improves in-distribution accuracy and out-of-distribution (OOD) robustness on ImageNet benchmarks, while SymphonySoftMoE achieves higher Top-5 accuracy and robustness. For multimodal tasks, SymphonySMoE outperforms the baseline in visual instruction tuning (LLaVA-332K), with consistent gains on ScienceQA, POPE, and MMBench.

## 5 EMPIRICAL ANALYSIS

We study models trained for the WikiText-103 language modeling task in this section.

**Shared Experts, More Experts, and More Experts Chosen.** We conduct additional experiments comparing SymphonySMoE to vanilla SMoE under two configurations: (A) 16 experts + top-2 routing + 1 additional shared expert and (B) 32 experts + top-4 routing. Here, in our shared expert setting, i.e., setting (A), the shared expert is set up similarly to that in DeepSeek-V2 (Liu et al., 2024a) and DeepSeek-V3 (Liu et al., 2024b). At each SMoE layer in (A), each token select the shared expert and use top-2 routing to select 2 other experts among the 16 experts. As shown in Table 4, SymphonySMoE consistently outperforms SMoE across all settings in both standard and robust perplexity metrics. Notably, the robustness gains are more pronounced in the larger configuration with more experts: SymphonySMoE reduces robust test perplexity from 45.72 to 40.07 in the 32 experts + top-4 routing, compared to a smaller drop from 44.19 to 42.79 in the 16 experts + top-2 routing presented in Table 1. This demonstrates SymphonySMoE's strong scalability and robustness as model capacity increases.

**Efficiency analysis.** We analyze the overhead introduced by Symphony routing compared to standard SMoE. The Symphony Router adds an $M \times M$ adjacency matrix and a lightweight update operation. As shown in Table 5, test-time computation involves a single $M \times M$ and $M \times N$

matrix multiplication, yielding a complexity of $\mathcal{O}(NM^2)$. Training adds an update step with complexity $\mathcal{O}\left(NC_2^k\right)$, but this component requires no gradient computation. Memory overhead remains low: $\mathcal{O}(M^2)$ at test time and

Table 5: Computational/Memory complexity introduced by the social graph in the Symphony Router

|  | Test Time | Training Time |
|---|---|---|
| **Computation** | $O(NM^2)$ | $O(NC_2^k + NM^2)$ |
| **Memory** | $O(M^2)$ | $O(NC_2^k + M^2)$ |

$\mathcal{O}\left(NC_2^k\right)$ during training. In typical cases, e.g., $M = 16$, $k = 2$, $N = 512$ as in Swin-MoE (Liu et al., 2021), this cost is negligible.

As shown in Tables 11–12 and Appendix F, SymphonySMoE incurs negligible overhead ($< 1\%$, diminishing with sequence length), with minimal additional cost even when integrated into large models like DeepSeek-V3.

**Additional Analysis.** Appendix G provides additional analyses, including social graph visualizations, ablations on initialization and moving average, load balancing, and dense vs. sparse graphs.

## 6 RELATED WORK

**Routing Mechanism.** Recent work has explored diverse token-expert routing strategies, including reinforcement learning (Bengio et al., 2015), deterministic hashing (Roller et al., 2021), optimal transport (Tianlin Liu, 2023), linear programming (Lewis et al., 2021), cosine similarity (Chi et al., 2022), soft token mixing (Puigcerver et al., 2024), greedy top-k expert selection per token (Shazeer et al., 2017a), and greedy top-k token selection per expert (Zhou et al., 2022c). Our Symphony Router complements existing approaches by improving token routing via expert-to-expert interactions. Moreover, its modular design enables seamless integration into a wide range of SMoE architectures.

**Robust SMoE.** Robustness in SMoE architectures is a growing research focus. (Puigcerver et al., 2022) analyzed model capacity and Lipschitz bounds for provable robustness, while (Zhang et al., 2023) proposed adversarial training for routers and experts. In contrast, our SymphonySMoE improves robustness by promoting the co-selection of expert pairs with high-confidence activation regions, as discussed in Remark 7.

**Graph and SMoE.** Graph-based approaches have recently been integrated into SMoE. (Nguyen et al., 2025) integrates token-to-token interactions into SMoE by formulating a probabilistic graphical model (PGM) that captures the structure of the token graph. While both (Nguyen et al., 2025) and our work utilize PGMs to incorporate structured interactions into SMoE, the two approaches are fundamentally different. Specifically, in contrast to (Nguyen et al., 2025), SymphonySMoE focuses on modeling expert-to-expert interactions rather than token-to-token interactions. Moreover, the PGM used in SymphonySMoE (Figure 1B) to capture expert-to-expert interactions is structurally distinct from the PGM proposed in (Nguyen et al., 2025) for capturing token-to-token interactions. We also emphasize that our PGM design in Figure 1B builds upon the classical MoE graphical model in (Xu et al., 1994) (Figure 1A), a foundational work in the literature on MoE (Yuksel et al., 2012). We provide more detailed comparisons with (Nguyen et al., 2025) in Appendix H.

Among the other works on graph and SMoE (Liu et al., 2023b; Kim et al., 2023; Zhang et al., 2024), (Wang et al., 2024a) applies MoE to GNNs for structural adaptability and efficiency. (Hu et al., 2022; Zhou et al., 2022b) address imbalance and generalization via MoE-based strategies, while (Tang et al., 2025) introduces a self-rethinking mechanism using pseudo-graph MoEs.

## 7 CONCLUDING REMARKS

In this paper, we propose SymphonySMoE, a novel class of SMoE that leverages the social graph between experts to enhance the token routing in the model. We derive SymphonySMoE and the construction of the expert-to-expert social graph from a posterior inference in a new probabilistic graphical model that we develop. We also theoretically prove that SymphonySMoE reinforces expert selection with the largest regions of ideal co-selection, enhancing the model's robustness in the presence of data contamination. We empirically validate the advantages of our SymphonySMoE over the baseline SMoE on a variety of popular benchmarks and practical tasks, including WikiText-103 language modeling, visual instruction tuning, and GLUE finetuning. A potential limitation of SymphonySMoE is that it only considers the experts within a single SMoE layer. Modeling the interactions between experts across different SMoE layers is an interesting research direction. It is also intriguing to further study the connections between our social graph construction in Algorithm 1 and the Hebbian learning rule as pointed out in Remark 3. We leave these exciting research directions as future work.

**Ethics Statement.** Given the nature of the work, we do not foresee any negative societal and ethical impacts of our work.

**Reproducibility Statement.** Source codes for our experiments are provided in the supplementary materials of the paper. The details of our experimental settings and computational infrastructure are given in Section 4 and the Appendix. All datasets that we used in the paper are published, and they are easy to access in the Internet.

**LLM Usage Declaration.** We use large language models (LLMs) for grammar checking and correction.

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

# Supplement to "Modeling Expert Interactions in Sparse Mixture of Experts via Graph Structures"

**Table of Contents**

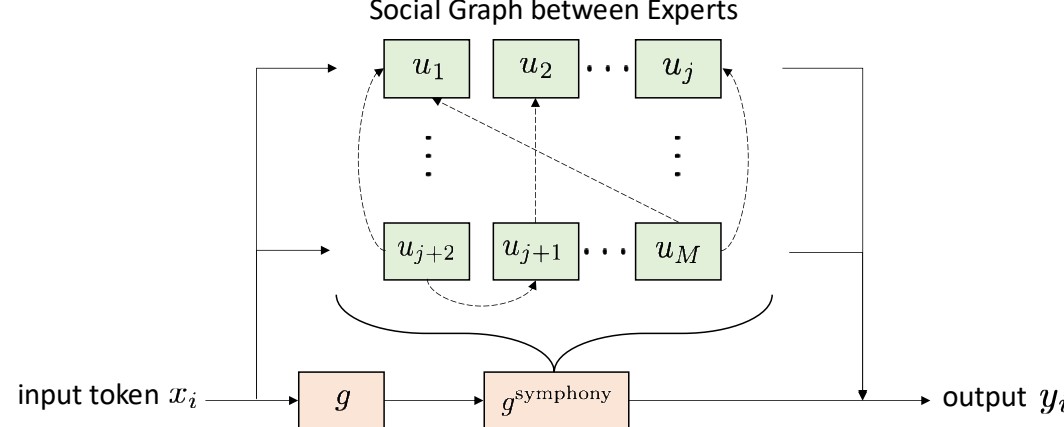

Figure 2: An overview figure that illustrates the architecture of SymphonySMoE.

# A OVERVIEW OF SYMPHONY SPARSE MIXTURE OF EXPERTS ARCHITECTURE

Figure 2 depicts the overall architecture of SymphonySMoE.

# B PROOFS

## B.1 PROOF OF THEOREM 1

*Proof of Theorem 1.* Let $\{\boldsymbol{w}_j\}_{j=1,\dots,M}$ be the expert embeddings. We can assume that the true selection region for expert $\boldsymbol{w}_j$ is well-estimated by a ball $\mathcal{B}_j = \mathcal{B}_j(R_j) := \{\boldsymbol{x} \in \mathcal{X} : \|\boldsymbol{x} - \boldsymbol{w}_j\| \leq R_j\}$ of radius $R_j > 0$ for all $j = 1, \dots, M$ since $\boldsymbol{w}_j^\top \boldsymbol{x} + b \approx -\frac{1}{2}\|\boldsymbol{w}_j - \boldsymbol{x}\|^2$ when the learnable bias is close to $-\frac{1}{2}(\|\boldsymbol{w}_j\|^2 + \|\boldsymbol{x}\|^2)$ (also note that $\arg\max_j(\boldsymbol{w}_j^\top \boldsymbol{x} + b) = \arg\min_j \|\boldsymbol{w}_j - \boldsymbol{x}\|$ holds exactly particularly when $\boldsymbol{w}_j$ are normalized as in Chi et al. (2022)), and Euclidean distance is spherically invariant as $\|\boldsymbol{w}_j - \boldsymbol{x}\| = \text{const}$ yields a sphere over $\boldsymbol{x}$. Then, the co-selection region for experts $\boldsymbol{w}_j$ and $\boldsymbol{w}_k$ can be represented as $\mathcal{C}_{jk} := \mathcal{B}_j \cap \mathcal{B}_k$.

Now consider the following contamination setting:

$$\mathcal{X}_\varepsilon := \left\{\boldsymbol{x}' \in \mathbb{R}^d : \boldsymbol{x}' = \boldsymbol{x} + \boldsymbol{\delta}, \ \boldsymbol{x} \in \mathcal{X}, \ \|\boldsymbol{\delta}\| \leq \varepsilon\right\}, \tag{9}$$

where $\mathcal{X}_\varepsilon$ essentially corresponds to the $\varepsilon$-expansion of $\mathcal{X}$. Subsequently, we define the following set of "escaped" tokens

$$\mathcal{C}_{jk}^{\text{esc}} := \left\{\boldsymbol{x} \in \mathcal{C}_{jk} : \boldsymbol{x} + \boldsymbol{\delta} \in \mathcal{X}_\varepsilon \setminus \mathcal{C}_{jk}, \ \|\boldsymbol{\delta}\| \leq \varepsilon\right\}, \tag{10}$$

which denotes all points in $\mathcal{C}_{jk}$ that are drifted out of $\mathcal{C}_{jk}$ because of natural data corruption or an adversary. Now since $\mathcal{C}_{jk}^{\text{esc}} \cup (\mathcal{C}_{jk} \setminus \mathcal{C}_{jk}^{\text{esc}}) = \mathcal{C}_{jk}$ with an empty overlap, we have

$$\frac{\mu(\mathcal{C}_{jk}^{\text{esc}})}{\mu(\mathcal{C}_{jk} \setminus \mathcal{C}_{jk}^{\text{esc}})} = \frac{\mu(\mathcal{C}_{jk}^{\text{esc}})}{\mu(\mathcal{C}_{jk}) - \mu(\mathcal{C}_{jk}^{\text{esc}})} \leq \frac{\mu(\mathcal{C}_{jk}^\varepsilon) - \mu(\mathcal{C}_{jk})}{\mu(\mathcal{C}_{jk}) - \left[\mu(\mathcal{C}_{jk}^\varepsilon) - \mu(\mathcal{C}_{jk})\right]}, \tag{11}$$

where $\mathcal{C}_{jk}^\varepsilon := \mathcal{B}_j(R_j + \varepsilon) \cap \mathcal{B}_k(R_k + \varepsilon)$, and the inequality follows from the fact that $\mathcal{C}_{jk}^{\text{esc}} \subseteq \mathcal{C}_{jk}^\varepsilon \setminus \mathcal{C}_{jk}$ and $\mathcal{C}_{jk} \subseteq \mathcal{C}_{jk}^\varepsilon$ for any $\varepsilon \geq 0$. Since measure (volume) of the intersection of balls is smooth in radius, we also have that $\lim_{\varepsilon \to 0^+} \mu(\mathcal{C}_{jk}^\varepsilon) = \mu(\mathcal{C}_{jk})$. Hence, for any small $\nu > 0$, there exists $\varepsilon_0 > 0$ such that for all $\varepsilon \leq \varepsilon_0$, the following holds:

$$\left|\mu(\mathcal{C}_{jk}^\varepsilon) - \mu(\mathcal{C}_{jk})\right| = \mu(\mathcal{C}_{jk}^\varepsilon) - \mu(\mathcal{C}_{jk}) \leq \frac{\nu}{1+\nu}\mu(\mathcal{C}_{jk}). \tag{12}$$

Plugging this into Eqn. 11, we obtain that

$$\frac{\mu(\mathcal{C}_{jk}^{\text{esc}})}{\mu(\mathcal{C}_{jk} \setminus \mathcal{C}_{jk}^{\text{esc}})} \leq \frac{\mu(\mathcal{C}_{jk}^\varepsilon) - \mu(\mathcal{C}_{jk})}{\mu(\mathcal{C}_{jk}) - \left[\mu(\mathcal{C}_{jk}^\varepsilon) - \mu(\mathcal{C}_{jk})\right]} \leq \frac{\frac{\nu}{1+\nu}\mu(\mathcal{C}_{jk})}{\mu(\mathcal{C}_{jk}) - \frac{\nu}{1+\nu}\mu(\mathcal{C}_{jk})} = \nu. \tag{13}$$

In other words, Eqn. 13 implies that the mass of the set of correctly routed tokens even under contamination of reasonably bounded magnitude, $\mu(\mathcal{C}_{jk} \setminus \mathcal{C}_{jk}^{\text{esc}})$, dominates the mass of the set of mis-routed tokens given by $\mathcal{C}_{jk}^{\text{esc}}$.

Notice also that the smoothness of volume as a function of radius yields $\mu(\mathcal{C}_{jk}^{\varepsilon}) - \mu(\mathcal{C}_{jk}) = \mu'(\mathcal{C}_{jk})\varepsilon + o(\varepsilon)$ where $\mu'(\mathcal{C}_{jk})$ is proportional to the surface of the intersection solid. Therefore, we have $\nu = O(\varepsilon)$ i.e. the rate of change in measure is of the same order as the perturbation magnitude.

Now observe that $a_{jk}$ is effectively an empirical estimation of the mass of tokens in $C_{jk} \setminus \mathcal{C}_{jk}^{\text{esc}}$ as, by definition,

$$a_{jk} = \frac{1}{N}\sum_{i=1}^{N}\mathbb{I}_{\boldsymbol{x}_i+\boldsymbol{\delta}\in\mathcal{C}_{jk}} \approx \mathbb{E}\left[\mathbb{I}_{\boldsymbol{x}_i\in(\mathcal{C}_{jk}\setminus\mathcal{C}_{jk}^{\text{esc}})\cup\mathcal{E}^{\text{esc}}}\right] = \mu((\mathcal{C}_{jk}\setminus\mathcal{C}_{jk}^{\text{esc}})\cup\mathcal{E}^{\text{esc}}) \leq \mu(\mathcal{C}_{jk}\setminus\mathcal{C}^{\text{esc}}) + L\varepsilon, \tag{14}$$

where $\mathcal{E}^{\text{esc}} \subseteq \bigcup_{j',k':|j-j'|+|k-k'|>0}\mathcal{C}_{j'k'}^{\text{esc}}$ denotes the set of tokens escaped from $\mathcal{C}_{j'k'}^{\text{esc}}$ and incoming into $\mathcal{C}_{jk}$ because of perturbation. The approximation holds for sufficiently large $N$ due to the Strong Law of Large Numbers (SLLN) while the last equality follows from the definition of expectation. The existence of such constant $L$ is provided by applying Eqn. 13 to all $\mathcal{C}_{j'k'}$ individually and using union bound. On the other hand, $\mathcal{C}_{jk}^{\text{esc}} \subseteq \mathcal{C}_{jk}^{\varepsilon} \setminus \mathcal{C}_{jk}$ and Eqn. 12 give that

$$\begin{aligned}
\mu(\mathcal{C}_{jk} \setminus \mathcal{C}_{jk}^{\text{esc}}) &= \mu(\mathcal{C}_{jk}) - \mu(\mathcal{C}_{jk}^{\text{esc}}) \\
&\geq \mu(\mathcal{C}_{jk}) - \mu(\mathcal{C}_{jk}^{\varepsilon} \setminus \mathcal{C}_{jk}) \\
&\geq \mu(\mathcal{C}_{jk}) - \frac{\nu}{1+\nu}\mu(\mathcal{C}_{jk}) \\
&= \frac{1}{1+\nu}\mu(\mathcal{C}_{jk}).
\end{aligned} \tag{15}$$

Thus, combining Eqn. 14 and Eqn. 15, we get

$$\frac{1}{1+\nu}\mu(\mathcal{C}_{jk}) \leq \mathbb{E}\left[\mathbb{I}_{\boldsymbol{x}_i\in(\mathcal{C}_{jk}\setminus\mathcal{C}_{jk}^{\text{esc}})\cup\mathcal{E}^{\text{esc}}}\right] \leq \mu(\mathcal{C}_{jk}) + L\varepsilon, \tag{16}$$

The only piece left is to find how good the approximation in Eqn. 14 is. To this end, since the indicator random variable is bounded between 0 and 1 by definition, Hoeffding's inequality (or McDiarmid's inequality when $\boldsymbol{\delta}$ is specifically a data-dependent adversarial perturbation to get a similar bound) gives that for any $t \geq 0$,

$$\mathbb{P}\left(\left|\frac{1}{N}\sum_{i=1}^{N}\mathbb{I}_{\boldsymbol{x}_i+\boldsymbol{\delta}\in\mathcal{C}_{jk}} - \mathbb{E}\left[\mathbb{I}_{\boldsymbol{x}_i\in(\mathcal{C}_{jk}\setminus\mathcal{C}_{jk}^{\text{esc}})\cup\mathcal{E}^{\text{esc}}}\right]\right| \geq t\right) \leq \exp(-2Nt^2). \tag{17}$$

Put differently, the following bound holds with probability $\geq 1 - \alpha$

$$\left|a_{jk} - \mathbb{E}\left[\mathbb{I}_{\boldsymbol{x}_i\in(\mathcal{C}_{jk}\setminus\mathcal{C}_{jk}^{\text{esc}})\cup\mathcal{E}^{\text{esc}}}\right]\right| \leq \sqrt{\frac{\ln(2/\alpha)}{2N}} \tag{18}$$

Combining all pieces together, with probability at least $1 - \alpha$, the following inequalities are true:

$$\frac{1}{1+\nu}\mu(\mathcal{C}_{jk}) - \sqrt{\frac{\ln(2/\alpha)}{2N}} \leq a_{jk} \leq \mu(\mathcal{C}_{jk}) + \sqrt{\frac{\ln(2/\alpha)}{2N}} + L\varepsilon. \tag{19}$$

Finally, notice that since $\nu = O(\varepsilon)$, we have $1/(1+\nu) = 1 - O(\varepsilon)$. Hence, there exists a constant $\tilde{L}$ such that

$$\mu(\mathcal{C}_{jk}) - \sqrt{\frac{\ln(2/\alpha)}{2N}} - \tilde{L}\varepsilon \leq a_{jk} \leq \mu(\mathcal{C}_{jk}) + \sqrt{\frac{\ln(2/\alpha)}{2N}} + \tilde{L}\varepsilon, \tag{20}$$

as claimed. $\square$

## B.2 RELAXING THE GEOMETRICAL ASSUMPTION ON EXPERT SELECTION REGIONS

While Theorem 1 assumes spherical expert selection regions for analytical tractability, the theoretical framework extends naturally to the polyhedral geometry inherent in sparse mixture-of-experts routing mechanisms. In practice, the router assigns token $\boldsymbol{x}$ to expert $j$ when $\boldsymbol{w}_j^{\top}\boldsymbol{x} + b_j \geq \boldsymbol{w}_k^{\top}\boldsymbol{x} + b_k$ for all $k \neq j$, defining Voronoi cells $\mathcal{V}_j := \{\boldsymbol{x} \in \mathbb{R}^d : \boldsymbol{w}_j^{\top}\boldsymbol{x} + b_j \geq \boldsymbol{w}_k^{\top}\boldsymbol{x} + b_k, \forall k \neq j\}$. The co-selection region becomes $\mathcal{C}_{jk} = \mathcal{V}_j^{(2)} \cap \mathcal{V}_k^{(2)}$, where $\mathcal{V}_j^{(2)}$ denotes the second-order Voronoi cell containing points for which expert $j$ ranks among the top-2 selections.

The key modification required is in the perturbation analysis. For polyhedral regions, the $\varepsilon$-expansion is given by the Minkowski sum $\mathcal{C}_{jk}^{\varepsilon} = \mathcal{C}_{jk} \oplus B_{\varepsilon}(0)$, where $B_{\varepsilon}(0)$ is the $\varepsilon$-ball centered at the origin. Following the Minkowski-Steiner formula for convex polytopes, the volume expansion can be expressed as:

$$\mu(\mathcal{C}_{jk}^{\varepsilon}) - \mu(\mathcal{C}_{jk}) = \sum_{i=1}^{d-1} V_{d-i}(\mathcal{C}_{jk}) \binom{d}{i} \varepsilon^i, \tag{21}$$

where the continuous functions $V_i(\mathcal{C}_{jk})$ denotes the $k$-th mixed volume (quermassintegral) of polytope $\mathcal{C}_{jk}$. In particular, $V_d(\mathcal{C}_{jk}) = \mu(\mathcal{C}_{jk})$ is the volume, $V_{d-1}(\mathcal{C}_{jk})$ is proportional to the surface area, and $V_1(\mathcal{C}_{jk})$ is the mean width.

For the linear term in $\varepsilon$, we have:

$$V_{d-1}(\mathcal{C}_{jk}) = \frac{1}{d} \sum_{F \in \mathcal{F}_{jk}} \sigma_{d-1}(F), \tag{22}$$

where $\mathcal{F}_{jk}$ denotes the set of $(d-1)$-dimensional faces of polytope $\mathcal{C}_{jk}$ and $\sigma_{d-1}(F)$ is the $(d-1)$-dimensional Hausdorff measure of face $F$. This replaces the spherical surface measure in the original analysis. The constant $\tilde{L}$ in Theorem 1 must be redefined to account for the polyhedral geometry:

$$\tilde{L} \propto \max_{j,k} \left\{ V_{d-1}(\mathcal{C}_{jk}) + \sum_{i=2}^{d-1} V_{d-i}(\mathcal{C}_{jk}) \binom{d}{i} \varepsilon_0^{i-1} \right\}, \tag{23}$$

where $\varepsilon_0$ is the maximum perturbation magnitude under consideration. The additional terms capture the contribution of higher-order geometric features (edges, vertices) to the expansion.

To ensure the bound in Theorem 1 remains valid, we require that the polyhedral regions $\mathcal{C}_{jk}$ satisfy a *geometric regularity condition*: there exists a constant $\rho > 0$ such that for all $j, k$, the polytope $\mathcal{C}_{jk}$ contains a ball of radius $\rho \cdot \mathrm{diam}(\mathcal{C}_{jk})$, where $\mathrm{diam}(\cdot)$ denotes the diameter. This condition prevents the occurrence of extremely thin or degenerate polytopes that could lead to unbounded sensitivity to perturbations.

Under this regularity condition, the polyhedral setting exhibits *anisotropic perturbation sensitivity*: tokens near faces of $\mathcal{C}_{jk}$ experience perturbation effects proportional to their distance from the hyperplane boundary, while tokens near edges and vertices face amplified sensitivity due to the local convergence of multiple decision boundaries. This geometric relaxation reinforces the intuition from Remark 7 that $a_{jk}$ naturally downweights expert pairs whose co-selection regions $\mathcal{C}_{jk}$ contain a higher density of boundary-proximal, contamination-vulnerable tokens.

### B.3  Proof of Proposition 1

*Proof.* We prove them in order.

(i) First, we decompose $s$ into two additive terms. Any probability vector $s \in \Delta^{M-1}$ (where $\mathbf{1}^{\top} s = 1$ and $s_j \geq 0$) can be decomposed as $s = u + v$, where $u = \frac{1}{M} \mathbf{1}$ is the "baseline" uniform vector ($\mathbf{1}$ is the all-ones vector) and $v$ is a perturbation satisfying $\mathbf{1}^{\top} v = 0$ (mean-zero condition).

Second, we examine the action of $A$ on $u$ and $v$ separately. Since $A$ is column-stochastic, $A\mathbf{1} = \mathbf{1}$, and thus $Au = A\left(\frac{1}{M}\mathbf{1}\right) = \frac{1}{M}\mathbf{1} = u$. For the perturbation $v$, we use the spectral gap assumption. The eigenvalues of $A$ are ordered as $1 = \lambda_1 > |\lambda_2| \geq \cdots \geq |\lambda_M|$, with $\rho = \max_{i \geq 2} |\lambda_i| < 1$. The spectral gap ensures that for any vector $v$ orthogonal to $\mathbf{1}$ (i.e., in the subspace spanned by the eigenvectors corresponding to $\lambda_i$, $i \geq 2$), $\|Av\|_2 \leq \rho \|v\|_2$. This follows from the fact that $Av$ is a linear combination of these eigenvectors, scaled by eigenvalues less than or equal to $\rho$ in magnitude:

$$\|Av\|_2^2 = \left( \sum_{i=2}^{M} \lambda_i \alpha_i \boldsymbol{\xi}_i \right)^{\top} \left( \sum_{i=2}^{M} \lambda_i \alpha_i \boldsymbol{\xi}_i \right) = \sum_{i=2}^{M} \lambda_i^2 \alpha_i^2 \leq \rho^2 \sum_{i=2}^{M} \alpha^2 = \rho^2 \|v\|_2^2,$$

where $\boldsymbol{\xi}_i$ are eigenvectors of $A$ and $\alpha_i$ are such that $v = \sum_{i=2}^{M} \alpha_i \boldsymbol{\xi}_i$.

Now, since $u^\top v = \left(\frac{1}{M}\mathbf{1}^\top\right)v = \frac{1}{M}(\mathbf{1}^\top v) = 0$, we have $\|s\|_2^2 = \|u\|_2^2 + \|v\|_2^2$. Similarly,

$$r = As = A(u + v) = Au + Av = u + Av,$$

which implies

$$\|r\|_2^2 = \|u + Av\|_2^2 = \|u\|_2^2 + \|Av\|_2^2 + 2u^\top Av.$$

Again, $u^\top Av = \left(\frac{1}{M}\mathbf{1}^\top\right)Av = \frac{1}{M}(\mathbf{1}^\top Av) = \frac{1}{M}(A^\top\mathbf{1})^\top v$. Since $A^\top\mathbf{1} = \mathbf{1}$ (as $A$ is column-stochastic, and symmetrization makes it doubly-stochastic), $u^\top Av = 0$. Thus,

$$\|r\|_2^2 = \|u\|_2^2 + \|Av\|_2^2 \leq \|u\|_2^2 + \rho^2\|v\|_2^2 \leq \|s\|_2^2,$$

so $\|r\|_2 \leq \|s\|_2$ holds as an intermediate result. For two probability vectors $s, s'$, write $s' - s = v'$ with $\mathbf{1}^\top v' = 0$, so

$$\|A(s' - s)\|_2 = \|Av'\|_2 \leq \rho\|v'\|_2 = \rho\|s' - s\|_2,$$

proving the contraction property. This is vital for robustness because mean-zero perturbations (e.g., noise that does not shift the total probability) are naturally damped, ensuring the routing remains stable against input fluctuations.

(ii) Consider a perturbation $\Delta s$ with $\|\Delta s\|_2 < g/(2\rho)$, where $g$ is the margin. Then $r' = A(s + \Delta s) = r + A\Delta s$. The $\ell_2$ norm bound gives $\|A\Delta s\|_2 \leq \rho\|\Delta s\|_2 < \rho \cdot \frac{g}{2\rho} = g/2$. In the $\ell_\infty$ norm, coordinate-wise changes satisfy $\|r' - r\|_\infty \leq \|A\Delta s\|_2 \leq g/2$. The margin $g$ ensures that the smallest value in $J$ exceeds the largest outside $J$ by $g$, so a change of less than $g/2$ cannot swap the TopK set. If $s(x)$ is $L_s$-Lipschitz, $\|\Delta s\|_2 \leq L_s\|\Delta x\|$, so $L_s\|\Delta x\| < g/(2\rho)$ implies $\|\Delta x\| < g/(2\rho L_s)$, completing the proof.

$\square$

## C   DERIVING COSINE AND RANDOM ROUTERS FROM THE POSTERIOR FORMULATION OF THE GATE VALUES IN (S)MoE

In this appendix, we show that the formulation of gate values in SMoE as a posterior distribution can be trivially applied to derive the cosine router (Chi et al., 2022) and random router (Chen et al., 2023).

**Deriving Random Router (Chen et al., 2023):** To derive the random router in (Chen et al., 2023), we just need to randomly initialized the center $\boldsymbol{w}_j$ of the likelihood $p(\boldsymbol{x}|z_j = 1) = \mathcal{N}(\boldsymbol{x}|\boldsymbol{w}_j, \sigma_j^2\mathbf{I})$ and the bias $b_j$ in Eqn. 5. We fix these parameters to guide token assignment during training.

**Deriving Cosine Router (Chi et al., 2022):** To derive the cosine router in (Chi et al., 2022), we first replace the router weight $\boldsymbol{W} := [\boldsymbol{w}_1, \ldots, \boldsymbol{w}_M]^\top \in \mathbb{R}^{M \times D}$ by another lower-dimensional router weight $\boldsymbol{E} := [\boldsymbol{e}_1, \ldots, \boldsymbol{e}_M]^\top \in \mathbb{R}^{M \times D_e}$. We next replace the token $\boldsymbol{x}$ by its projected representation onto the lower-dimensional expert embedding space $\Omega\boldsymbol{x}$ where $\Omega \in \mathbb{R}^{D_e \times D}$. We normalize $\Omega\boldsymbol{x}$ and $\boldsymbol{e}_i, i = 1, \ldots, M$. Similar derivation as in Section 2.1 can then be applied to obtain the gate values of the cosine router.

## D   EXPERIMENTAL DETAILS

### D.1   WIKITEXT-103 LANGUAGE MODELING

**Dataset.** The WikiText-103 dataset (Merity et al., 2017) is derived from Wikipedia articles and is designed to capture long-range contextual dependencies. The training set contains about 28,000 articles, with a total of 103 million words. Each article is divided into text blocks with approximately 3,600 words. The validation and test sets have 218,000 and 246,000 words, respectively, with both sets comprising 60 articles and totaling about 268,000 words. On the attacked dataset, we corrupt the both validation and test data to demonstrate the robustness of SymphonySMoE using TextAttack's word swap attack (Morris et al., 2020). This adversarial attack randomly replaces words in the dataset with a generic "AAA" for evaluation making it difficult for the model to predict the next word in the sequence correctly.

**Model and baselines.** We use the Switch Transformer (Fedus et al., 2022), referred to as SMoE in our tables and figures, and GLaM (Du et al., 2022) baselines, which replaces each multilayer perceptron (MLP) layer and every other MLP layer in a vanilla language modeling transformer with a SMoE layer, respectively.

For consistency, we define the number of layers in each model as the number of SMoE layers. The default model used in each experiment is medium sized with 6 layers, but we include a comparison between a larger one with 12 layers as well. Each model has 16 experts in every SMoE layer and selects 2 experts ($K = 2$) per input. The baseline SMoE model uses a sparse router function consisting of a linear network receiving the input data followed by the TopK, then the Softmax function. The SymphonySMoE model replaces that linear network with our Symphony router. The medium and large SMoE models train for 80 epochs and the GLaM models train for 120 epochs. Our implementation is based on the code base developed by (Press et al., 2020), publicly available at https://github.com/ofirpress/sandwich_transformer and https://github.com/giangdip2410/CompeteSMoE/tree/main.

### D.2    IMAGENET-1K OBJECT RECOGNITION

**Datasets.** We use the ImageNet-1K dataset that contains 1.28M training images and 50K validation images. There are 1000 classes of images and the model learns an object recognition task. For robustness to common corruptions, we use ImageNet-C (IN-C) (Hendrycks & Dietterich, 2019b) which consists of 15 different types of corruptions applied to the ImageNet-1K validation set with 5 levels of severity. To test robustness to label distribution shifts, we use ImageNet-O (Hendrycks et al., 2021c). This dataset contains a 200-class subset of ImageNet-1K classes with adversarially filtered images. Finally, we test our model on ImageNet-R (IN-R) (Hendrycks et al., 2021b), which contains various artistic renditions of images. This evaluates the model's generalization ability to abstract visual renditions.

**Metrics.** On ImageNet-1K and ImageNet-R, we report the top-1 accuracies for all experiments. On ImageNet-C, the standard metric for evaluation is the mCE. To calculate this, we average the top-1 error rate for each corruption type across the 5 levels of severity and divide them by AlexNet's average errors, then take the final average across all corruption types. We report the area under the precision-recall curve (AUPR) for ImageNet-O, which requires anomaly scores. The score is obtained by taking the negative of the highest softmax probability output by the model. The direction of increasing or decreasing values of these metrics, signifying greater robustness, will be indicated in the table with an arrow.

**Model and baselines.** The baseline model we use is the SMoE version of the Swin Transformer (Liu et al., 2021) architecture which has a total of 280M parameters. This backbone uses 4 base layers of depth 2, 2, 18, and 2. The first two base layers each contain 2 self-attention layers and 2 feed-forward layers. The third base layer contains 18 self-attention layers with alternating feed-forward and SMoE layers. The final base layer contains 2 self-attention layers with one feed-forward and one MoE layer. The embedding dimension is 96 and the heads per base layer are 3, 6, 12, and 24. We use a total of 16 experts per SMoE layer use top-2 expert routing. We train both Swin-MoE model and SymphonySwin-MoE models for 60 epochs using the publicly available code, https://github.com/microsoft/Swin-Transformer?tab=readme-ov-file.

### D.3    VISUAL INSTRUCTION TUNING

**Training pipeline.** We adapt the training pipeline in CuMO (Li et al., 2024) to train a Mixture of Experts (MoE) model from existing dense Large Language Model (LLM) and Visual Model checkpoints using the Sparse Upcycling technique (Komatsuzaki et al., 2022). This approach duplicates the original model to create experts and continues training them on a downstream dataset as a normal MoE, bypassing the expensive pre-training step. The pipeline follows a two-stage process: (1) **Dense Training Stage**, where the MLP connector is initialized and trained to connect the pre-trained visual encoder to the pre-trained LLM using the LLaVA-558K dataset (Liu et al., 2024d); (2) **MoE Training Stage**, where the model is upcycled to become MoE and trained on the full LLaVA-665K dataset to obtain visual instruction-following capabilities. We inherit the pretrained dense checkpoint of Siglip224 + Phi3.5 model and the training code from LibMoE (Nguyen et al., 2024).

**Evaluation benchmarks.** The GQA (Hudson & Manning, 2019) and MMMU (Yue et al., 2024) benchmarks assess the model's visual perception capabilities for tasks ranging widely from day-to-day tasks to college-level tests. Following LLaVA-1.5 (Liu et al., 2024c), ScienceQA (Lu et al., 2022) with multiple-choice questions is used to evaluate zero-shot generalization in scientific question answering. TextVQA (Singh et al., 2019) and MMStar (Chen et al., 2024) further test the model's text and visual capabilities through text-rich visual questions and visually indispensable questions. Lastly,

Table 6: Top-1 accuracy (%) of Swin-MoE and SymphonySwin-MoE on each corruption type in ImageNet-C for the largest severity level.

| Corruption Type/Model | Swin-MoE *(Baseline)* | SymphonySwin-MoE (Ours) |
|---|---|---|
| Brightness | 58.49 | 59.16 |
| Contrast | 22.48 | 21.75 |
| Defocus Blur | 13.92 | 14.35 |
| Elastic Transform | 26.84 | 27.26 |
| Fog | 32.82 | 28.88 |
| Frost | 36.59 | 38.04 |
| Gaussian Noise | 14.10 | 16.69 |
| Glass Blur | 10.79 | 11.67 |
| Impulse Noise | 13.68 | 17.60 |
| JPEG Compression | 17.10 | 18.84 |
| Motion Blur | 21.47 | 21.68 |
| Pixelate | 8.58 | 10.18 |
| Shot Noise | 13.42 | 16.21 |
| Snow | 31.43 | 33.41 |
| Zoom Blur | 23.57 | 23.60 |

Table 7: Perplexity (PPL) results of SymphonySMoE vs. SMoE baseline on clean and attacked C4 dataset subset validation and test sets. A lower PPL implies better performance.

| Model | Clean C4 subset | | Attacked C4 subset | |
|---|---|---|---|---|
| | Valid PPL ↓ | Test PPL ↓ | Valid PPL ↓ | Test PPL ↓ |
| SMoE (baseline) | 40.38 | 40.55 | 51.24 | 51.34 |
| SymphonySMoE (Ours) | **39.47** | **39.82** | **49.78** | **50.26** |

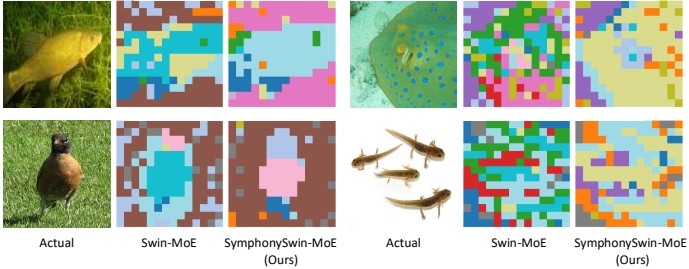

Figure 3: Plots of 14x14 image reconstructions where each patch is colored by its assigned expert in the Swin-MoE (baseline) and SymphonySwin-MoE model. For each triple of images, the left-most image is the actual input to each model, the middle image is a visualization of the assigned expert for each patch in the baseline Swin-MoE while the right-most image is the corresponding plot for the SymphonySwin-MoE model.

we use MMBench (Liu et al., 2024e) with comprehensive multiple-choice questions and POPE (Li et al., 2023b) with object hallucination tests to evaluate the model's robustness.

# E  ADDITIONAL EXPERIMENTAL RESULTS

## E.1  C4 LANGUAGE MODELING

We trained a Switch Transformer (Fedus et al., 2022) and its Symphony counterpart on 8B unique tokens from C4, following the setup of (Muennighoff et al., 2023). For validation and test sets, we used 80M tokens each, and constructed attacked versions using the method described in Appendix D.1. Both models were trained for a single epoch, with the best checkpoints selected based on validation perplexity. As shown in Table 7, SymphonySMoE outperforms the vanilla SMoE on both clean and attacked data. On clean C4, SymphonySMoE improves test perplexity by +0.73 (39.82 vs. 40.55). Under attack, it achieves a larger gain of +1.08 (50.26 vs. 51.34), demonstrating greater robustness. Together, these C4 results highlight the effectiveness of our method even under single-epoch training on very large corpora, a more realistic setting than multi-epoch training.

Table 8: Top-1 accuracy (%), top-5 accuracy (%), AUPR and mean corruption error (mCE) of SymphonySMoE vs. the Swin-MoE baseline on ImageNet-1K and popular robustness benchmarks for image classification. For all metrics except mCE, a higher value is better. For mCE on ImageNet-C, lower is better.

| Metric/Model | | *Swin-MoE (baseline)* | SymphonySwin-MoE (Ours) |
|---|---|---|---|
| Parameters | | 280 M | 280 M |
| IN-1K | Top-1 ↑ | 75.39 | **75.57** |
| IN-R | Top-1 ↑ | 31.50 | **31.72** |
| IN-O | AUPR ↑ | 18.36 | **18.52** |
| IN-C | mCE ↓ | 74.28 | **73.56** |

### E.2 IMAGENET-1K OBJECT RECOGNITION ON SWINMOE

**Experimental Setup.** We follow the experimental setup of (Liu et al., 2021) for pre-training and evaluation on ImageNet-1K (IN-1K). In particular, we evaluate SymphonySwin-MoE against the Swin-MoE Transformer baseline with a total of 16 experts and top-2 expert routing.

**Out-of-distribution Image Classification.** We conducted additional experiments on the standard out-of-distribution (OOD) benchmarks for robustness in computer vision tasks. These robustness benchmarks include image classification on ImageNet-R (IN-R) (Hendrycks et al., 2021a), ImageNet-O (IN-O) (Hendrycks et al., 2021c), and ImageNet-C (IN-C) (Hendrycks & Dietterich, 2019a). We provide details on each dataset and the metrics for evaluation in Appendix D.2. As ImageNet-C consists of 15 different corruption types across 5 levels of severity, we provide the average result in Table 8 and the full set of results in Appendix E.3. We adopt the conventional setup of pre-training on the clean ImageNet-1K dataset and evaluating the trained models on the various OOD datasets (Han et al., 2024; Zhou et al., 2022a; Puigcerver et al., 2022).

On clean data, our SymphonySwin-MoE achieves an 0.2% increase in top-1 accuracy while on ImageNet-R, ImageNet-O and ImageNet-C, we also observe good improvements. Most notably, SymphonySwin-MoE achieves a substantial boost in performance on ImageNet-C, reducing the baseline's result by more than 0.7 mCE. Combining these results with those on the language modeling task in Section 4.1, we illustrate the effectiveness of our Symphony Router across multiple modalities, further highlighting the strength of our approach.

**Image Segmentation.** In each example in Figure 3, we observe that SymphonySwin-MoE is able to identify distinct, meaningful regions more effectively than the baseline model and route these regions to the same experts. The improved segmentation of the images allows each expert to better specialize in specific features in the data, leading to improved performance on clean ImageNet-1K.

### E.3 FULL SET OF IMAGENET-C RESULTS

In this section, we present the top-1 accuracy of Swin-MoE and SymphonySwin-MoE on each corruption type in ImageNet-C for the largest severity level. These results can be found in Table 6. Across almost all corruption categories, we see a remarkable improvement in the accuracy of the SymphonySwin-MoE. In particular, under the Impulse Noise corruption, we gain almost 4% in top-1 accuracy.

### E.4 IMAGENET-1K OBJECT RECOGNITION ON SOFTMOE

To further demonstrate the integrability of our method on vision tasks, we evaluate a SoftMoE variant with the symphony router integrated, referred to as SymphonySoftMoE. The experiments follow the same setup described in Section E.2. As shown in Table 9, SymphonySoftMoE achieves improvements in Top-5 accuracy on the clean ImageNet-1K test set and consistently enhances robustness on ImageNet-R and ImageNet-O.

### E.5 LLAVA-332K RESULTS

**Results on LLaVA-332K dataset.** In visual instruction tuning task, we finetune both SymphonySMoE and SMoE on LLaVA-332K to further testing our approach in the limited data scenario. The result in Table 10 shows the advantage of SymphonySMoE when it performs better than the baseline SMoE in all benchmarks, especially with remarkably margin of 2.47% on the ScienceQA benchmarks.

Table 9: Top-1 and top-5 accuracy on clean ImageNet-1K (IN-1K), top-1 accuracy on ImageNet-R (IN-R) and AUPR on ImageNet-O (IN-O) for the baseline SoftMoE model and SymphonySoftMoE. For all metrics, higher is better.

| Metric/Model | | *SoftMoE* *(baseline)* | SymphonySoftMoE (Ours) |
|---|---|---|---|
| IN-1K | Top-1 ↑ | **72.21** | 72.14 |
| | Top-5 ↑ | 90.57 | **90.59** |
| IN-R | Top-1 ↑ | 32.41 | **32.56** |
| IN-O | AUPR ↑ | 17.59 | **17.84** |

Table 10: Accuracy of SMoE and SymphonySMoE in Visual Instruction Tuning task. Both models is upcycled from Siglip224 + Phi3.5 with a total of 4.2B parameters and finetuned on LLaVA-665K (Liu et al., 2024c) dataset. We evaluate the model performance across 7 popular benchmarks with diverse characteristics, especially hallucination (POPE (Li et al., 2023b)) and robustness (MMBench (Liu et al., 2024e)). A higher accuracy indicates better performance.

| Benchmark | SMoE (baseline) | SymphonySMoE (Ours) |
|---|---|---|
| TextVQA | 39.33 | **40.42** |
| GQA | 59.16 | **60.03** |
| MMMU | 41.89 | **43.22** |
| MMStar | 41.67 | **42.27** |
| ScienceQA | 79.56 | **82.03** |
| MMBench-EN | 70.19 | **70.53** |
| POPE | 85.22 | **85.91** |

## F COMPLEXITY AND RUNTIME ANALYSIS

In this section, we examine both the theoretical computational complexity and the empirical runtime performance of SymphonySMoE, demonstrating that it introduces only a minimal computational overhead.

### F.1 COMPLEXITY OF THE SYMPHONY ROUTER

From a theoretical standpoint, the Symphony Router defined in Section 2 introduces only an $M \times M$ adjacency matrix and a corresponding update operation. Table 5 outlines the additional computational and memory complexities incurred by the router. Computationally, this design entails just one $M \times M$ and one $M \times N$ matrix multiplication, resulting in a test-time complexity of $O(NM^2)$. During training, an extra update operation is required with a complexity of $O(NC_2^k)$, where $N$ denotes the number of tokens. Importantly, this component does not require gradient computation, which simplifies the training process. In terms of memory, the overhead introduced by the social graph remains modest: storing the symphony adjacency matrix at test time requires only $O(M^2)$ memory, while training necessitates $O(NC_2^k)$ memory to maintain the expert-expert pair list. In practical scenarios, the number of experts $M$ and the top-$k$ value $k$ are typically much smaller than the sequence length $N$.

As a concrete example, we analyze the theoretical overhead of DeepSeek-V3 (Liu et al., 2024b), a large-scale Mixture-of-Experts (MoE) model composed of 58 MoE layers ($L$), each containing 256 routed experts ($M$), with a top-$k$ selection of 8 ($k$), and a context length of 4096 tokens ($N$). The additional memory costs during training and inference are calculated as $L(M^2 + NC_k^2) \times 4B$ and $LM^2 \times 4B$, yielding approximately 39.875 MB and 14.5 MB, respectively. These overheads are negligible compared to the 1250 GB model size, and thus have no significant impact on storage or deployment. On the computational side, the training and inference overheads are $LN(M^2 + C_k^2)$ and $LNM^2$ floating point operations, respectively, amounting to 14.51G and 14.5G FLOPs. These values remain small relative to the hundreds of tera-FLOPs required by DeepSeek-V3 and the 19.5 TFLOPS FP32 processing capability of an Nvidia A100 GPU (NVIDIA Corporation, 2020).

### F.2 RUNTIME EVOLUTION OF SYMPHONYSMoE

**Evaluation Settings.** We use the base Switch-Transformer settings in D.1 in WikiText-103 Language Modeling task and conduct the experiments on a single A100-SXM4-80GB. For the runtime and memory evolution at test time, we fix batch size $B = 8$, topK $k = 2$ and vary the number of experts $M \in \{4, 16, 32, 64\}$, the sequence length $N \in \{256, 512, 1024, 2048, 4096\}$.

Table 11: Runtime comparison between SMoE and SymphonySMoE at test time (time in ms)

| #experts | 4 | | | 16 | | | 32 | | | 64 | | |
|---|---|---|---|---|---|---|---|---|---|---|---|---|
| seqlen | SMoE | SymphonySMoE | Δ(%) | SMoE | SymphonySMoE | Δ(%) | SMoE | SymphonySMoE | Δ(%) | SMoE | SymphonySMoE | Δ(%) |
| 256 | 26.72 | 26.81 | +0.35 | 27.19 | 27.29 | +0.37 | 29.37 | 29.72 | +1.19 | 33.98 | 34.17 | +0.56 |
| 512 | 44.27 | 44.25 | -0.06 | 44.47 | 44.85 | +0.86 | 45.05 | 45.14 | +0.21 | 49.55 | 49.64 | +0.19 |
| 1024 | 81.10 | 81.20 | +0.13 | 81.86 | 82.09 | +0.28 | 81.99 | 82.26 | +0.33 | 82.77 | 83.23 | +0.55 |
| 2048 | 164.64 | 165.28 | +0.39 | 165.74 | 166.35 | +0.37 | 167.46 | 168.07 | +0.36 | 168.17 | 168.75 | +0.34 |
| 4096 | 411.92 | 412.29 | +0.09 | 413.52 | 413.35 | -0.04 | 414.36 | 415.17 | +0.20 | 417.59 | 418.55 | +0.23 |

Table 12: Memory usage comparison between SMoE and SymphonySMoE at test time (memory in MB)

| #experts | 4 | | | 16 | | | 32 | | | 64 | | |
|---|---|---|---|---|---|---|---|---|---|---|---|---|
| seqlen | SMoE | SymphonySMoE | Δ(%) | SMoE | SymphonySMoE | Δ(%) | SMoE | SymphonySMoE | Δ(%) | SMoE | SymphonySMoE | Δ(%) |
| 256 | 5624 | 5624 | 0.00 | 5706 | 5714 | +0.14 | 5798 | 5806 | +0.14 | 5990 | 5998 | +0.13 |
| 512 | 10164 | 10164 | 0.00 | 10252 | 10252 | 0.00 | 10336 | 10336 | 0.00 | 10518 | 10518 | 0.00 |
| 1024 | 19504 | 19504 | 0.00 | 19600 | 19600 | 0.00 | 19684 | 19686 | +0.01 | 19884 | 19884 | 0.00 |
| 2048 | 39130 | 39130 | 0.00 | 39226 | 39228 | +0.01 | 39310 | 39310 | 0.00 | 39510 | 39510 | 0.00 |
| 4096 | 73160 | 73162 | +0.003 | 73256 | 73256 | 0.00 | 73340 | 73340 | 0.00 | 73542 | 73542 | 0.00 |

Table 13: Perplexity of SymphonySMoE and SMoE under different initialization strategies on clean and attacked WikiText-103 validation and test sets. A lower PPL implies better performance.

| Experiment Name | Clean WikiText-103 | | Attacked WikiText-103 | |
|---|---|---|---|---|
| | Valid PPL ↓ | Test PPL ↓ | Valid PPL ↓ | Test PPL ↓ |
| *SMoE baseline* | 33.76 | 35.55 | 42.24 | 44.19 |
| Uniform SymphonySMoE | 33.50 | 35.40 | 41.89 | 44.12 |
| Kaiming SymphonySMoE | 32.77 | 34.34 | 41.09 | 42.99 |
| Xavier SymphonySMoE | 32.97 | 34.32 | 41.36 | 42.95 |
| Zero SymphonySMoE | **32.71** | **34.29** | **40.87** | **42.79** |

**Runtime and memory evolution at test time.** We include the result for runtime evolution at Table 11 and for memory evolution at Table 12.

### F.3 REDUCING OVERHEAD

We have employed a fully vectorized implementation of all operations introduced by SMoE, ensuring that no manual loops are introduced and that all computations are GPU-friendly. The code for this implementation is provided in the supplementary material.

Note that SymphonySMoE framework requires invoking the TopK operation twice per sample: once to construct the expert-expert adjacency matrix $A$ and once to compute the final scores. However, the first invocation, which updates $A$, does not participate in gradient computation and thus does not interfere with the optimization process.

## G ADDITIONAL EMPIRICAL ANALYSIS

### G.1 ABLATION STUDY ON DIFFERENT INITIALIZATION STRATEGIES FOR THE SOCIAL GRAPH

The social graph is represented by the adjacency matrix $A \in \mathbb{R}^{M \times M}$, where $M$ denotes the number of experts. At initialization, this matrix is set to the zero matrix. We name this the zero initialization (Zero SymphonySMoE).

We conduct additional experiments with three alternative initialization strategies: Kaiming, Xavier, and standard uniform initialization. The results of these experiments are reported in Table 13. As shown in the results, these alternative initialization methods improve both the accuracy and robustness of SymphonySMoE compared to the baseline model. However, the zero initialization strategy remains the most effective among the tested approaches.

### G.2 ABLATION STUDY ON THE DECAY PARAMETER $\beta$

The parameter $\beta$ serves as the decay factor in the exponential moving average used to interpolate between the current and new expert-to-expert interaction graphs during training. We conducted an ablation study on $\beta$, with results summarized in Table 14 below. Across all tested values, the model consistently outperforms the baseline, though the degree of improvement varies.

The best performance is achieved at $\beta = 0.9$, suggesting that retaining more historical information about expert interactions is beneficial for both clean and robust performance. This indicates that

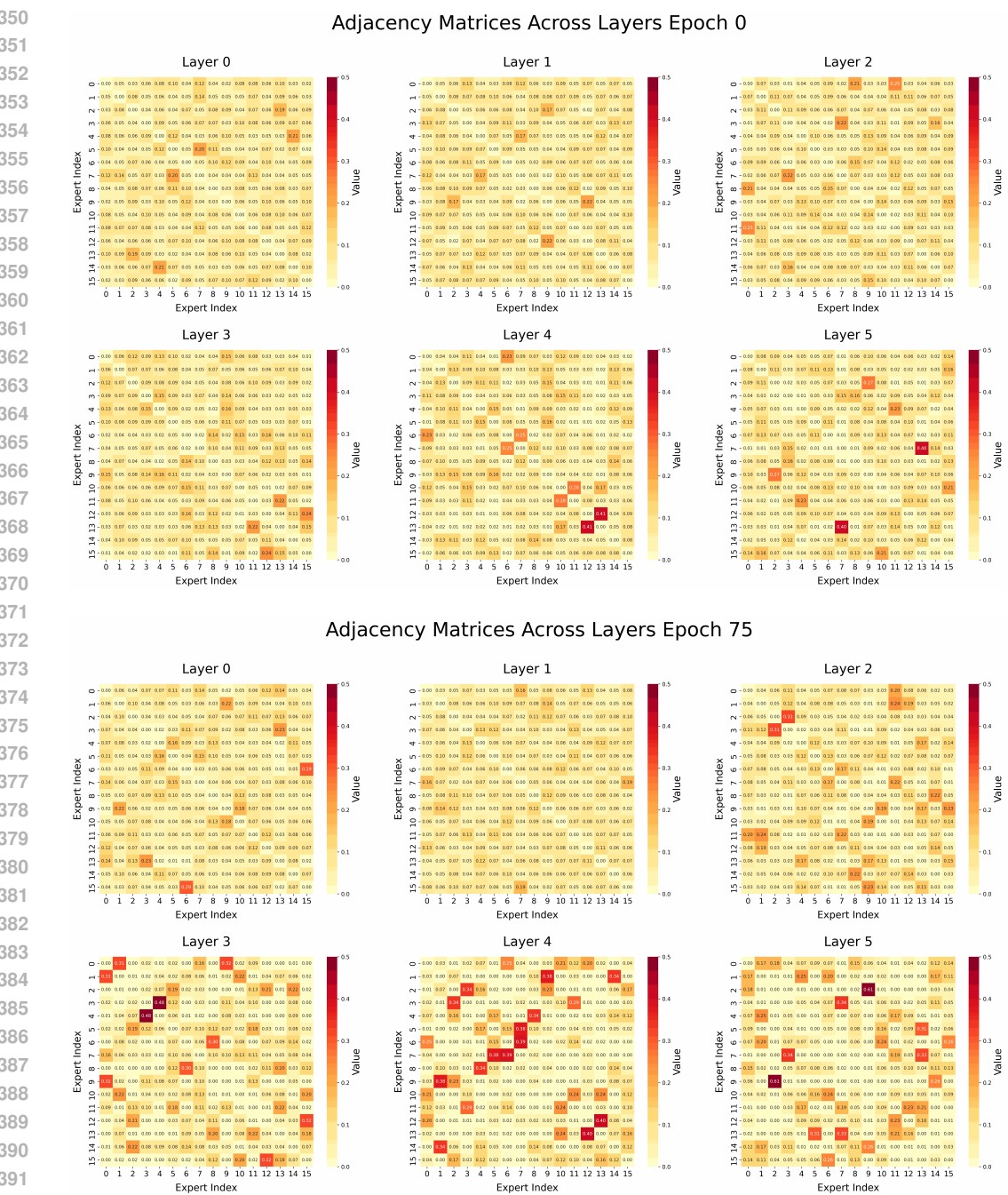

Figure 4: Evolution of social graphs across layers in Switch Transformers during Wikitext-103 pretraining.

expert specialization patterns evolve gradually, and maintaining a longer-term memory of these relationships helps stabilize routing decisions. Importantly, the consistent improvements across a broad range of $\beta$ values (from 0.3 to 0.9) demonstrate that our method is robust to this hyperparameter, which enhances its practical applicability.

## G.3 DENSE VS. SPARSE SOCIAL GRAPH

To further explore potential reductions in overhead, we experimented with sparsifying the social graph using a threshold $t = 0.05$. The results of this experiment are reported in Table 15. While introducing sparsity still improves performance compared to the baseline model, it leads to a decrease in both accuracy and robustness of SymphonySMoE. Additionally, given that the matrix size is $M^2$ (which is $16 \times 16$ in our case), the difference in memory and computational cost between the dense

Table 14: Ablation study on decay parameter $\beta$ on clean and attacked WikiText-103 validation and test sets. A lower PPL implies better performance.

| $\beta$ | Clean WikiText-103 | | Attacked WikiText-103 | |
|---|---|---|---|---|
| | Valid PPL ↓ | Test PPL ↓ | Valid PPL ↓ | Test PPL ↓ |
| baseline SMoE | 33.76 | 35.55 | 42.24 | 44.19 |
| $\beta = 0.9$ | **32.71** | **34.29** | **40.87** | **42.79** |
| $\beta = 0.7$ | 33.24 | 35.2 | 41.78 | 44.1 |
| $\beta = 0.5$ | 32.89 | 34.41 | 41.33 | 43.06 |
| $\beta = 0.3$ | 33.03 | 34.38 | 41.33 | 42.97 |

Table 15: Perplexity of SymphonySMoE with sparse graph initialization. We evaluate the computational efficiency and performance impact of sparsifying the social graph threshold t = 0.05. Sparsity leads to a decline in accuracy and robustness. Lower perplexity indicates better performance.

| Experiment Name | Clean WikiText-103 | | Attacked WikiText-103 | |
|---|---|---|---|---|
| | Valid PPL ↓ | Test PPL ↓ | Valid PPL ↓ | Test PPL ↓ |
| *SMoE (baseline)* | 33.76 | 35.55 | 42.24 | 44.19 |
| Sparse SymphonySMoE | 33.00 | 34.53 | 41.13 | 42.88 |
| **Dense SymphonySMoE** | **32.71** | **34.29** | **40.87** | **42.79** |

and sparse representations is negligible. Consequently, we opted to retain the dense formulation to preserve model performance.

### G.4 LOAD BALANCING

**Load balancing.** Figure 5 compares expert load balancing between the baseline SMoE and our proposed SymphonySMoE on WikiText-103 task. Experts in each layer are ordered by their selection frequency during validation, and these frequencies are averaged across all SMoE layers. The left plot shows a right-skewed distribution in the baseline model, indicating a load imbalance where a few experts dominate. In contrast, SymphonySMoE (right plot) exhibits a more uniform distribution, suggesting significantly improved distribution, suggesting significantly improved

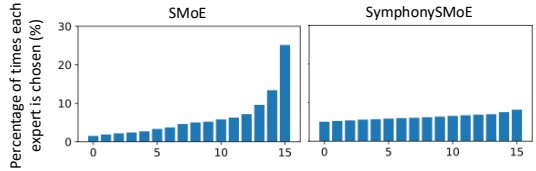

Figure 5: Expert selection frequencies for baseline SMoE and SymphonySMoE on WikiText-103, averaged across all layers. Experts are sorted by frequency; a more uniform distribution reflects better load balancing.

load balancing that leads to better model performance. These findings align with the improved results on WikiText-103 reported in Table 1.

### G.5 VISUALIZING THE SOCIAL GRAPH

Figure 4 visualizes the evolution of social graphs across all layers of Switch Transformers trained on WikiText-103. The results show progressive refinement in expert usage, with deeper layers exhibiting stronger, more structured connections, hinting for the emerging expert specialization and collaboration dynamics in the training.

## H DISTIGUISH WITH OTHER GRAPH-BASED SMoE

We briefly compared our SymphonySMoE model with (Nguyen et al., 2025) in the Related Work section of the manuscript. Here, we present the key differences between SymphonySMoE and the Similarity/Attention-Aware SMoE proposed by (Nguyen et al., 2025).

**Key Conceptual Difference** The fundamental distinction lies in the type of structural interaction leveraged for routing in sparse mixture-of-experts (SMoE) models. SymphonySMoE introduces and exploits *expert-to-expert interactions* to guide token routing, whereas Similarity/Attention-Aware SMoE incorporates *token-to-token interactions* into the routing mechanism. These two approaches are complementary rather than mutually exclusive. To demonstrate this, we conducted additional experiments, reported in Table 16. Integrating SymphonySMoE's expert-to-expert interactions into Similarity-Aware SMoE improves the latter's performance in both clean and noisy (attacked) settings (e.g., PPL improves from $34.50 \rightarrow 35.06$ under clean and $42.83 \rightarrow 43.67$ under noise). Interestingly, SymphonySMoE alone still outperforms both Similarity-Aware SMoE and the combined model. This

Table 16: Perplexity (PPL) results of SymphonySMoE vs. SMoE baseline and similarity-informed variants on clean and attacked WikiText-103 validation and test sets. A lower PPL implies better performance.

| Model/Metric | Clean WikiText-103 | | Attacked WikiText-103 | |
|---|---|---|---|---|
| | Valid PPL ↓ | Test PPL ↓ | Valid PPL ↓ | Test PPL ↓ |
| *SMoE (baseline)* | 33.76 | 35.55 | 42.24 | 44.19 |
| Similarity-inform SMoE | 33.31 | 35.06 | 41.59 | 43.67 |
| Symphony + Similarity-inform SMoE | 32.92 | 34.50 | 41.22 | 42.83 |
| SymphonySMoE (Ours) | **32.71** | **34.29** | **40.87** | **42.79** |

Table 17: Perplexity (PPL) results of 32 experts-top 4 setting of SymphonySMoE vs. SMoE baseline and similarity-informed variants on clean and attacked WikiText-103 validation and test sets. A lower PPL implies better performance.

| Model/Metric | Clean WikiText-103 | | Attacked WikiText-103 | |
|---|---|---|---|---|
| | Valid PPL ↓ | Test PPL ↓ | Valid PPL ↓ | Test PPL ↓ |
| *SMoE 32-4 (baseline)* | 30.77 | 32.71 | 43.77 | 45.72 |
| Similarity-inform SMoE 32-4 | 30.87 | 32.58 | 39.26 | 41.20 |
| SymphonySMoE 32-4 (Ours) | **30.54** | **32.01** | **38.29** | **40.07** |

Table 18: Throughput (samples/second) of SymphonySMoE vs. SMoE baseline and Similarity-informed SMoE during training and inference on the WikiText-103 dataset. We also compare the memory consumption of these models at inference time. Higher throughput and lower memory consumption indicate better efficiency.

| Model/Metric | Train throughput (samples/second) ↑ | Inference throughput (samples/second) ↑ | Inference memory (MB) ↓ |
|---|---|---|---|
| *SMoE (baseline)* | **1066** | **3677** | **66319** |
| Similarity-inform SMoE | 951 | 3433 | 70077 |
| SymphonySMoE (Ours) | 1006 | 3603 | 68929 |

suggests that effectively fusing expert- and token-level graphs for routing remains an open challenge and a promising direction for future research.

Furthermore, the probabilistic graphical model (PGM) used in SymphonySMoE (Figure 1B) to capture expert-to-expert interactions is structurally distinct from the PGM in (Nguyen et al., 2025), which models token-to-token interactions. Our design (Figure 1B) builds directly on the classical MoE graphical model of (Xu et al., 1994) (Figure 1A), a foundational work in the MoE literature.

**Advantages of Expert-to-Expert Interactions in SymphonySMoE** Beyond conceptual novelty, SymphonySMoE's expert-to-expert interactions provide two practical advantages over token-to-token approaches. First, they offer improved scalability with large expert pools. Recent SMoE-based models such as DeepSeekV3 (Liu et al., 2024b) employ large expert sets (e.g., one shared + 256 routed experts). As shown in Table 4 of our manuscript, SymphonySMoE scales favorably with the number of experts. In contrast, it remains unclear whether token-to-token approaches exhibit similar scalability. To further investigate, we compared SymphonySMoE and Similarity-Aware SMoE on WikiText-103 with 32 experts. Results in Table 17 show that as the number of experts grows, SymphonySMoE's robustness advantage under word-swap attacks increases (PPL gap rises from 0.88 to 1.13).

Second, SymphonySMoE achieves better computational and memory efficiency. It requires only an adjacency matrix of size $M \times M$ for $M$ experts, whereas Similarity-/Attention-Aware SMoE requires an interaction matrix of size $N \times N$ for $N$ tokens. The computational complexity of SymphonySMoE is $\mathcal{O}(NM^2)$, compared to $\mathcal{O}(N^2M)$ for token-based methods. In practical scenarios, $M \ll N$; for example, in the WikiText-103 experiments (Table 1), $M = 16$ while $N = 512$. Additionally, SymphonySMoE does not require recomputing the expert graph during inference, whereas token-to-token approaches recompute token graphs for each input sequence. Table 18 compares training/inference throughput (samples/sec) and memory usage (MB) between SymphonySMoE and Similarity-Aware SMoE on WikiText-103. The results confirm that SymphonySMoE achieves faster training and inference speeds and lower memory costs, while maintaining performance comparable to baseline SMoE.

## I  BROADER IMPACTS

Our research enhances both clean data handling and robust performance, particularly in socially impactful domains. Notably, we demonstrate improved results in object recognition, benefiting self-driving cars, and language modeling, enhancing AI chatbot assistants. We show significant advancements in resisting data perturbation, aiming to protect critical AI systems from malicious actors. Furthermore, we achieve competitive performance in language modeling with contaminated data, reflecting real-world scenarios where data is often imperfect. While the potential for AI misuse exists, our work provides substantial improvements in fundamental architectures and theory, which we hope will lead to further socially beneficial outcomes.

