# OpenReview forum: "Modeling Expert Interactions in Sparse Mixture of Experts via Graph Structures"
_ICLR.cc/2026/Conference — Submitted to ICLR 2026_

### Official Review · Reviewer_Tv3D · 2025-10-23

**Soundness:** 2
**Presentation:** 2
**Contribution:** 2
**Rating:** 4
**Confidence:** 4

**Summary:**

The paper proposes SymphonySMoE, an extension of Sparse Mixture of Experts (SMoE). The paper first views routing as a probabilistic graphical model. Next, it incorporates a social graph to model interactions between experts and adjusts the SMoE gating values accordingly. This approach seeks to enhance token routing by promoting the co-selection of expert pairs with high-confidence activations. The authors present a theoretical analysis and empirical evaluation across language modeling (WikiText-103), visual instruction tuning (LLaVA), and GLUE fine-tuning.

**Strengths:**

- The paper's primary strength is the introduction of a novel social graph framework for modeling expert-to-expert interactions within  SMoE system.
- The paper also presents a strong theoretical analysis that rigorously formalizes the co-selection properties of experts.
- Extensive empirical evaluation on several models and multiple domains.

**Weaknesses:**

- The paper's probabilistic graphical model is primarily conceptual; the practical method simply uses an adjacency matrix to smooth gating scores, with the PGM adding no material impact to the final routing.
- The paper does not benchmark against other recent advanced routing strategies (see [1] for a list of possible baselines).
- Across most benchmarks, the reported improvements are modest (e.g., 1–3% absolute in some multimodal tasks, ~0.5–1 perplexity drop in WikiText-103). Without a rigorous statistical significance analysis or evaluation on more challenging datasets, such as mathematical reasoning, it is difficult to conclude that the improvements are meaningful in practice.
- The GLUE experiments are limited to Phi3-SMoE with top-2 selection among 4 experts, which again does not stress test the method’s scalability to larger, more realistic SMoE architectures.
- While the adjacency matrix update is claimed to be lightweight, according to the complexity analysis in Table 5 for large N (e.g., long sequences) or large M, this could become non-negligible.

[1] Do et al. "On the Effectiveness of Discrete Representations in Sparse Mixture of Experts.", TMLR 2025.

**Questions:**

- How sensitive is the method to the way the adjacency matrix is constructed (e.g., co-activation frequency, normalization, smoothing)?
- What are the results with more experts in the GLUE benchmark?

---

### Official Review · Reviewer_gs8E · 2025-10-28

**Soundness:** 2
**Presentation:** 2
**Contribution:** 3
**Rating:** 2
**Confidence:** 4

**Summary:**

This paper introduces SymphonySMoE, a novel MoE routing mechanism that incorporates experts' co-selection information into the routing decision.
The authors frame the routing process in MoE as a graphical model and provide a theoretical analysis of their design.
They conduct experiments to demonstrate that this routing design enables MoE to adapt to distributional shifts, leading to more robust routing.
This approach presents an interesting perspective to MoE routing design, though further empirical validation and improved paper presentation would enhance the overall impact of the paper.

**Strengths:**

- The idea of using experts' co-occurrence information to provide a smoothing signal for MoE routing is novel. To the best of my knowledge, there is little prior work in this area, making it an interesting contribution.

- The authors provide a therotical analysis to support their idea.

**Weaknesses:**

- The authors' claim that SymphonySMoE addresses the distributional shifts in traditional MoEs lacks a logical foundation.
I do not see, nor can I understand, any motivation linking SymphonySMoE to this concept of robustness throughout the paper.
I understand that SymphonySMoE uses the mutual information between experts to help MoE routing, but the connection between this mechanism and robustness is unsubstantiated, lacking proper explanation and empirical validation.

-  Some of the author's claims are not adequately supported by experimental evidence, as the experiments suffer from significant setup issues.

I question the validity of the authors' claims (i), (ii), and (iv) in the overview of Section 4.

Regarding claim (i) that "SymphonySMoE enhances model performance across both pre-training and fine-tuning tasks," I have the following concerns:

(1) In Section 4.1, the authors train a MoE model with a total of 200M parameters from scratch on only **100M** tokens (WikiText-103) and report this as a **pre-training** task.
It is difficult to draw convincing conclusions from such a limited **100M** token **"pre-training"** experiment and believe it can justify a new MoE routing strategy.
Could the observed results simply be due to SMoE enabling faster convergence?

Furthermore, I cannot accept the results of a language model **without any pre-training** on the attacked dataset as sufficient evidence to support the claim that SymphonySMoE is more robust. (claim (ii))

(2) I do not consider the experiments in Sections 4.1 and 4.2 as fine-tuning tasks, as there is **no** pre-trained MoE model involved.
These experiments are conducted with MoE initialized from the upcycled dense model, without any further training.
Similar to point (1), I find it difficult to accept conclusions drawn from tuning a newly initialized MoE with billions of parameters on such a limited dataset.
As a result, this also fails to support the conclusions of claim (iv).

- The presentation of this paper could be improved. Most theoretical proofs in the main text do not focus on addressing the problem this paper try to resolve and could be moved to the appendix, while some key experimental details that support the effectiveness of the approach are placed there instead.

**Questions:**

Q1: Can the authors conduct experiments pre-training language models with more tokens and a bigger model scale?

For instance, pre-train the language model with 80B tokens, similar to the setup in Appendix E.1.

Q2: Can the authors provide an explanation for their choice to upcycle dense models into MoE in the experiments presented in Sections 4.2 and 4.3?
What's the performance of continual pre-training performance of an MoE model into SymphonySMoE in the same experimental setup?

Q3: What's the performance of the fine-tuned dense Phi-3 mini's performance on GLUE?

Could the authors consider testing on other benchmarks, as GLUE may not fully capture the capabilities of modern LLMs?
As a kind reminder, the statement "this setup reflects a realistic deployment scenario" seems somewhat overstated, as modern MoEs are clearly much sparser in practice.

---

### Official Review · Reviewer_kqhn · 2025-10-29

**Soundness:** 4
**Presentation:** 3
**Contribution:** 4
**Rating:** 8
**Confidence:** 3

**Summary:**

This paper proposes SymphonySMoE, a novel framework for improving the robustness and interpretability of Sparse Mixture of Experts (SMoE) models by explicitly modeling expert-to-expert interactions through a graph structure.
Traditional SMoE frameworks select top-K experts independently for each token, which leads to unstable routing under distribution shift or noisy inputs. SymphonySMoE addresses this by constructing a social graph among experts, where edges represent co-activation frequency. This graph is dynamically updated via exponential moving average and used to smooth routing logits during expert selection.
Experiments on large-scale benchmarks—including WikiText-103 (language modeling), GLUE (text classification), and LLaVA-665K (vision-language instruction tuning)—demonstrate consistent performance gains over strong SMoE baselines (e.g., X-MoE, GLaM, Switch Transformer), particularly under noisy or adversarial conditions.
The paper further provides theoretical analysis showing that the learned adjacency matrix converges to an ideal co-activation measure, explaining the enhanced routing stability.

**Strengths:**

The paper introduces a new perspective on SMoE routing, framing it as a graph-based probabilistic inference problem that models dependencies among experts, rather than treating expert activations as independent.
The concept of a “social graph of experts” is both intuitively appealing and technically original, bridging ideas from graph neural networks, probabilistic modeling, and mixture-of-experts learning.
It offers a lightweight and modular extension that can be integrated into existing SMoE frameworks with minimal architectural modification.
The method is mathematically well-motivated and empirically validated across multiple modalities (text, vision-language).
Experiments are comprehensive, ablation studies isolate the impact of graph modeling, and robustness tests under data corruption demonstrate practical benefits.
Theoretical analysis provides a convergence guarantee for the adjacency matrix, which strengthens the credibility of the approach.

**Weaknesses:**

The paper reports stable gains on multiple benchmarks (such as table results in the directions of WikiText-103, GLUE, and LLaVA), and conducts a detailed complexity/runtime analysis of the overhead, but does not characterize the theoretical or empirical upper limit of Symphony routing: How far is the current improvement from the "ideal route", under what conditions will it reach its peak, and where will the diminishing returns occur?

**Questions:**

1.How sensitive is model performance to the EMA decay rate in updating the adjacency matrix? Would a fully learnable adjacency (trained via gradient) perform better or risk overfitting?
2.The experiments show improvement on text and vision-language tasks—does the method generalize similarly to purely visual MoE models (e.g., ViT-MoE) or speech experts?
3.Could the authors provide quantitative metrics for “expert interaction strength” or visualize how the graph evolves across training stages? This might better substantiate the social-graph analogy.
4.Please answer Weaknesses.

---

### Official Review · Reviewer_ZsKP · 2025-10-30

**Soundness:** 3
**Presentation:** 2
**Contribution:** 2
**Rating:** 4
**Confidence:** 5

**Summary:**

This work enhances the expert routing process in SMoE by incorporating the experts co-activation information. The proposed method, SymphonySMoE, construct an a social graph (co-occurrence matrix) to model the co-activation frequency of experts during training and modify the routing process. The authors provided theoretical analysis of SymphonySMoE and validate its efficacy on several scenarios, from pre-training to fine tuning and visual instruction tuning.

**Strengths:**

- The idea of incorporating expert co-activation frequency is well-motivated.
- SymphonySMoE is quite elegant. Despite its simple implementation, it is theoretically-grounded and the empirical results are encouraging.

**Weaknesses:**

- My major concern of this work is the empirical evaluation is quite limited.
    - First, the pre-training experiment is very small. Training ~220M models on WikiText-103 is quite limited. Furthermore, evaluation is also on the same dataset, such in-domain evaluation is not used in modern SMoE settings, most of which focus on zero-shot evaluation. A minimum scale for pre-training should be MoEUT [A] or preferably OLMoE [B].
    - Second, finetuning Phi 3 on Glue seems to be unnecessary as it is a very old benchmark and Phi 3 is likely to see the data during its pre-training. For this experiment, it is mandatory to report the original Phi 3 performance, and also consider challenging benchmarks like SuperGlue. Preferably, the authors should consider finetuning on more recent datasets like OpenCodeInstruct [C], or even doing RLHF.
    - Lastly, the visual instruction tuning experiment followed LibMoE, which reported 11 benchmarks, why did the authors only consider 7?
- Some presentation/typos/citation errors at L128, L139, L156, L161, etc. Table 1 appears too early before it was first mentioned.

[A] Csordás, Róbert, et al. "Moeut: Mixture-of-experts universal transformers." Advances in Neural Information Processing Systems 37 (2024): 28589-28614.

[B] Muennighoff, Niklas, et al. "Olmoe: Open mixture-of-experts language models." arXiv preprint arXiv:2409.02060 (2024).

[C] Ahmad, Wasi Uddin, et al. "OpenCodeInstruct: A Large-scale Instruction Tuning Dataset for Code LLMs." arXiv preprint arXiv:2504.04030 (2025).

**Questions:**

- The empirical evaluation of SymphonySMoE is quite limited.

- It is nice to see that the overheads during evaluation is minimal. What is the wall clock training time of SymphonySMoE compared to the baselines?

- The number of baselines considered in all experiments is quite limited. The authors should try to include more recent baselines such as MoEUT, Autonomy-of-Experts Models [D], etc.

[D] Lv, Ang, et al. "Autonomy-of-Experts Models." ICML (2025).

---

### Author Response · Authors · 2025-12-02
**Message 1 for New AC - Summary of Our Key Contributions**

**Methodology.** In our paper, we propose a principled method for constructing a social graph that represents relationships among experts. We then introduce SymphonySMoE, a new gating mechanism that incorporates inter-expert interactions into the gate scores.

**Theory.** Based on strong foundational theory, we prove that, under mild assumptions, SymphonySMoE provides two major advantages compared with the vanilla counterpart: (1) it improves the accuracy and stability of the routing mechanism by promoting the co-selection of high-confidence expert pairs for a given input token, and (2) it improves routing robustness by contracting mean-zero perturbations and increasing the margin required for an adversarial Top-K change.

**Experiments.** Our extensive experimental results on both training and sparse upcycling across text and vision domains confirm these advantages and further show that our framework is lightweight and universal, meaning it can be incorporated into diverse gating mechanisms and is not limited to the standard softmax SMoE gating.

---

### Author Response · Authors · 2025-12-02
**Message 2 for New AC - Summary of Main Concerns of Reviewers and Our Replies**

The reviewers raised three main concerns. We summarize them and provide our responses below:

1. **Limited empirical evaluation in terms of pretraining scale and baselines.**

Regarding the baseline concern, we note that SymphonySMoE is evaluated against a diverse set of configurations, including the GLaM architecture, XMoE, SwinMoE, scaling with more experts, and DeepSeekMoE-style shared-expert techniques. Across these comparisons, we consistently observe substantial improvements in robustness alongside strong performance gains in both language and vision domains.

For the pretraining scale, our experiments are conducted in a multi-epoch setting on WikiText-103, a well-established benchmark in the MoE literature and widely adopted in recent works [1, 2, 3]. While Reviewer ZsKP suggests pretraining on the C4 dataset, a standard pretraining run would require approximately 110 GPU-hours on A100-80GB hardware per run [4]. As an academic research group, we operate under strict computational constraints and are therefore unable to support such large-scale pretraining experiments. We believe our current experimental setup, aligned with prior MoE studies, provides a reasonable and informative evaluation of SymphonySMoE under these practical limitations.

2. **The probabilistic graphical model is primarily conceptual and has limited impact on routing.**

In our paper, we review the SMoE from the perspective of probabilistic graphical models (PGM) [5]. In particular, the gate values from the MoE router can be expressed as the posterior distribution of a mixture model with a uniform prior, and the gate values in SMoE are simply its truncated approximation.

Building on this foundation, we introduce a new PGM that explicitly models expert-to-expert relationships within SMoE. From this formulation, we derive the Symphony Sparse Mixture of Experts (SymphonySMoE), a model that improves token routing by leveraging a learned “social graph” among experts, enabling more coordinated and structured interactions across the expert set.

This PGM-based perspective opens up rich opportunities to connect MoE architectures with established methodologies and theories in probabilistic graphical models, as well as related areas such as social network analysis and recommender systems. SymphonySMoE serves as a concrete demonstration of this approach's potential.

3. **The complexity introduced by the adjacency matrix could become non-negligible during training and inference.**

We provide an empirical overhead analysis for Switch Transformer in **Appendix F** of our manuscript and further complement this with a theoretical overhead analysis for DeepSeek-V3, a large-scale Mixture-of-Experts (MoE) model. DeepSeek-V3 consists of 58 MoE layers ($L$), each with 256 routed experts ($M$), top-$k$ selection with $k = 8$, and a context length of 4096 tokens ($N$).

During training and inference, the additional memory costs introduced by the adjacency matrix are $L(M^2 + N C_k^2) \times 4\text{B}$ and $L M^2 \times 4\text{B}$, respectively, which correspond to approximately **39.875 MB** and **14.5 MB**. Compared to the **1250 GB** model size of DeepSeek-V3, this additional memory overhead is negligible and does not affect storage or deployment.

From a computational standpoint, the additional training and inference costs are $L N (M^2 + C_k^2)$ and $L N M^2$, resulting in **14.51 G** and **14.5 G** floating-point operations, respectively. These overheads are small relative to the hundreds of teraFLOPs required for DeepSeek-V3 training, as well as the **19.5 TFLOPS** FP32 processing capability of a single NVIDIA A100 GPU.

   Overall, these analyses indicate that the computational and memory overhead introduced by the social graph remains minimal, even for extremely large MoE models.

**References**

[1] Yang, Yuanhang, et al. "XMoE: Sparse Models with Fine-grained and Adaptive Expert Selection." ACL, 2024.

[2] Chi, Zewen, et al. "On the representation collapse of sparse mixture of experts." NeurIPS, 2022.

[3] Zhao, Hao, et al. "HyperMoE: Towards Better Mixture of Experts via Transferring Among Experts." ACL, 2024.

[4] Csordás, Róbert, et al. "Moeut: Mixture-of-experts universal transformers." NeurIPS, 2024.

[5] Xu, Lei, et al. "An alternative model for mixtures of experts." NeurIPS, 1994.

---

### Author Response · Authors · 2025-12-04
**We Appreciate Your Willingness to Oversee Our Submission**

Dear AC,

We are grateful that you have stepped in at this point in the review cycle to assume responsibility for our submission. Your time and attention are sincerely appreciated.

To facilitate your evaluation of both the paper and its review history, we have organized two concise messages, which appear in the comments that follow:

Message 1. Summary of Novelties and Contributions of Our Work.

Message 2. Summary of Primary Reviewer Questions and Our Responses.

Most of the reviewers' concerns focus on the experimental setup of the model rather than on the methodology itself. One of the main points raised is that pretraining ~220M-parameter models on WikiText-103 is not sufficient, and that the experiments should instead involve larger-scale pretraining on the C4 dataset. We argue that pretraining on WikiText-103 is a common practice in MoE research, and many prior works have adopted this setting [1, 2, 3]. Another concern is the validity of the sparse-upcycling setting for demonstrating the fine-tuning performance of our method. We again contend that sparse upcycling is a standard practice for SMoE and has been used in many previous works [4, 5].

We believe that, beyond the numerical results, the core idea of leveraging a probabilistic graphical model framework for mixture-of-experts is itself valuable and worthy of recognition. Furthermore, we consider the experiments presented in the paper to be sufficiently extensive to demonstrate the advantages of SymphonySMoE. Specifically, we include three main experiments: pretraining on Wikitext-103; sparse upcycling of billion-parameter vision-language and language models on practical tasks (LLaVa, GLUE); and experiments on a subset of the C4 dataset, as well as pretraining on ImageNet. Our baselines are diverse and up to date, including XMoE, SoftMoE, Swin-MoE, and DeepSeek-style shared experts. We also provide an ablation study on the effect of graph initialization, hyperparameter choices, and an efficiency analysis. Therefore, we believe that the requested larger-scale experiments would yield diminishing returns. Moreover, as a research group with limited resources, we are unable to conduct experiments at that scale within the short time frame available.

Thank you once again for overseeing our submission. If any aspect of the paper or this rebuttal requires clarification, please do not hesitate to contact us. We would be glad to provide further details or engage in additional discussion.

Warm regards,

The Authors

[1] Yang, Yuanhang, et al. "XMoE: Sparse Models with Fine-grained and Adaptive Expert Selection." ACL, 2024.

[2] Chi, Zewen, et al. "On the representation collapse of sparse mixture of experts." NeurIPS, 2022.

[3] Zhao, Hao, et al. "HyperMoE: Towards Better Mixture of Experts via Transferring Among Experts." ACL, 2024.

[4] Li, Jiachen, et al. "Cumo: Scaling multimodal llm with co-upcycled mixture-of-experts." NeurIPS, 2024.

[5] Hu, Shengding, et al. "MiniCPM: Unveiling the Potential of Small Language Models with Scalable Training Strategies." COLM, 2024.

---

### Meta-Review · Area_Chair_aE8t · 2026-01-06

**Summary:**

Across reviews, there is agreement that the core idea of modeling expert-to-expert interactions for SMoE routing is interesting and conceptually well motivated, and that the paper makes a genuine effort to provide theoretical justification. However, reviewers raise consistent and substantial concerns about whether the empirical evidence is sufficient to support the paper’s claims at ICLR’s acceptance bar.

The dominant concern is the scope and scale of empirical validation. Multiple reviewers note that the main “pretraining” experiment is conducted at a relatively small scale (~200M parameters trained on WikiText-103), with in-domain evaluation only, which is not representative of modern MoE pretraining practices nor adequate to substantiate claims about robustness or general effectiveness. Relatedly, robustness claims are viewed as under-supported, since evaluations rely on limited corruption settings without large-scale pretraining or strong out-of-distribution benchmarks.

Reviewers also question the choice and modernity of benchmarks. GLUE is widely regarded as outdated for evaluating modern LLMs, especially for models such as Phi-3 that may have seen similar data during pretraining, and several reviewers suggest that stronger or more contemporary benchmarks would be necessary. For visual instruction tuning, reviewers note that benchmark coverage is narrower than prior protocols (e.g., LibMoE), reducing confidence in the generality of the reported gains.

In addition, several reviewers point out missing comparisons to more recent or competitive MoE routing baselines, and express concern that the probabilistic graphical model framing, while elegant, may be largely conceptual relative to the implemented adjacency-based smoothing mechanism.

Taken together, while the idea is promising and technically sound, reviewers do not find the current empirical evidence sufficiently strong or comprehensive to support the paper’s claims, leading to a recommendation below the acceptance threshold.

**Reviewer Concerns:**

It seems like no specific rebuttal was provided by the authors, and therefore none of the substantive reviewer concerns were addressed during the discussion phase. In particular, concerns regarding the scope and scale of empirical evaluation, benchmark selection, missing comparisons to recent MoE routing baselines remain outstanding. As a result, the reviewers’ original reservations continue to hold.

**Reviewer Scores:**

see above

---

### Decision · Program_Chairs · 2026-01-26

Reject